# Transforming Transformers for Resilient Lifelong Learning

## Abstract

Lifelong learning without catastrophic forgetting (i.e., resiliency) remains an open problem for deep neural networks. The prior art mostly focuses on convolutional neural networks. With the increasing dominance of Transformers in deep learning, it is a pressing need to study resilient lifelong learning with Transformers. Due to the complexity of training Transformers in practice, for lifelong learning, a question naturally arises: Can the Transformer be learned to grow in a task aware way, that is to be dynamically tranformed by introducing lightweight learnable plastic components to the architecture, while retaining the parameter-heavy, but stable components at streaming tasks? To that end, motivated by the lifelong learning capability maintained by the functionality of Hippocampi in human brain, this paper explores what would be, and how to implement, Artificial Hippocampi (ArtiHippo) in Transformers. It presents a method of identifying, and then learning to grow, ArtiHippo in Vision Transformers (ViTs) for resilient lifelong learning in four aspects: (i) Where to place ArtiHippo in ViTs to enable plasticity while preserving the core function of ViTs at streaming tasks? (ii) What representational scheme to use to realize ArtiHippo to ensure expressivity and adaptivity for tackling tasks of different nature in lifelong learning? (iii) How to learn to grow ArtiHippo to exploit task synergies (i.e., the learned knowledge) and to overcome catastrophic forgetting? (iv) How to harness the best of our proposed ArtiHippo and prompting-based approaches? In experiments, the proposed method is tested on the challenging Visual Domain Decathlon (VDD) benchmark and the recently proposed 5-Dataset benchmark under the task-incremental lifelong learning setting. It obtains consistently better performance than the prior art with sensible ArtiHippo learned continually. To our knowledge, it is the first attempt of lifelong learning with ViTs on the challenging VDD benchmark.

## 1 Introduction

Developing lifelong learning machines is one of the hallmarks of AI, to mimic human intelligence in terms of learning-to-learn to be adaptive and skilled at streaming tasks. However, state-of-the-art machine (deep) learning systems realized by Deep Neural Networks (DNNs) are not yet intelligent in the biological sense from the perspective of lifelong learning, especially plagued with the critical issue known as *catastrophic forgetting* at streaming tasks in a dynamic environment (McCloskey & Cohen, 1989; Thrun & Mitchell, 1995). Catastrophic forgetting means that these systems "forget" how to solve old tasks after sequentially and continually trained on a new task using the data of the new task only. Addressing catastrophic forgetting in lifelong learning is a pressing need with potential paradigm-shift impacts in the next wave of trustworthy and/or brain-inspired AI.

To address catastrophic forgetting, there are two main categories of lifelong learning methods: exemplar-based methods (Aljundi et al., 2019b; Hayes et al., 2019; Wu et al., 2019) and exemplar-free methods (Kirkpatrick et al., 2017; Li et al., 2019; Wang et al., 2022d;c;a), both of which have witnessed promising progress. The former is also referred to Experience Replay, where a small number of selected samples is stored, using either raw data or latent feature representations, for each previous task, and then incorporated in conjunction with the data of a new task in training the model for the new task. The key lies in how to select the so-called coreset of experience and how the experience replaying is actually performed. For the latter, since no data of previous tasks in any forms will be available, the focus is typically on how to retain the model parameters trained for

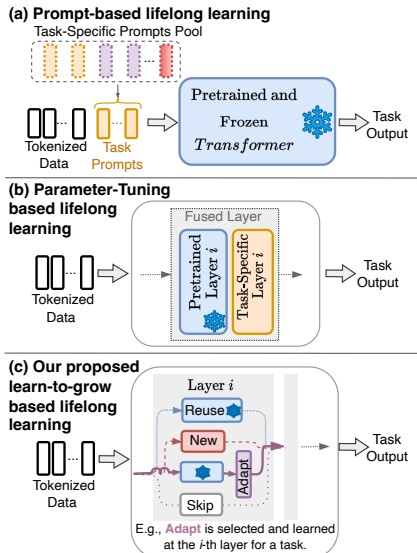

**(a) Prompt-based lifelong learning**

**(b) Parameter-Tuning based lifelong learning**

**(c) Our proposed learn-to-grow based lifelong learning**

Figure 1: Illustration of the difference between the prior art of lifelong learning using Transformers (Vaswani et al., 2017) and our proposed learn-to-grow method. *(a) Prompt-based methods* (Wang et al., 2022d;c;a) leverage a pretrained and frozen Transformer and learn task-specific prompts. *(b) Parameter-tuning based methods* introduce task-specific layer-wise parameters on top of a pretrained and frozen Transformer. One key lies in how the pretrained layer and the task-specific layer are fused, e.g., the parameter-masking methods (Wortsman et al., 2020; Xue et al., 2022; Mallya et al., 2018) and the output-addition methods (Ge et al., 2023b;a; Ermis et al., 2022; Morgado & Vasconcelos, 2019). Another key is where to apply the parameter tuning, which is often overlooked. *(c) Our proposed method* belongs to the parameter-tuning category with more fine-grained control. It utilizes four learn-to-grow operations: `Reuse`, `New`, `Adapt` and `Skip`, motivated by the learn-to-grow method (Li et al., 2019). We study where to place the learning-to-grow module in Transformers, what representational scheme to use, how to select the best among the four learn-to-grow operations, and how to integrate with prompt-based methods. See text for details.

previous tasks in training the model for a new task. There are three main strategies in adapting a current model for a new task. The first is to regularize the change in model parameters such as the popular Elastic Weight Consolidation (EWC) method (Kirkpatrick et al., 2017). The second is to dynamically expand the model such as the learn-to-grow method (Li et al., 2019), the Supermask in Superposition (SupSup) method (Wortsman et al., 2020), the Lightweight Learner (Ge et al., 2023b), the calibration method (Singh et al., 2020), the efficient feature transformation method (Verma et al., 2021) and the Channel-wise Lightweight Reprogramming (CLR) method (Ge et al., 2023a), all of which have been mainly developed for convolutional neural networks. More recently, with the availability of powerful pretrained Transformer (Vaswani et al., 2017) based Large Foundation Models (LFMs) (Bommasani et al., 2021) (such as the CLIP models (Radford et al., 2021)), the third is to freeze pretrained LFMs and then to learn prompts (or task tokens) appended to the input tokens instead for lifelong learning, i.e. prompting-based methods (Wang et al., 2022d;c;a).

With the increasing dominance of Transformers in deep learning, in this paper, we are interested in studying how to dynamically expand Transformers in a lightweight way for resilient lifelong learning, that is to develop an exemplar-free lifelong learning method for Transformers, which is complementary to the prompt-based approaches. Fig. 1 illustrates the proposed method which is not built on completely frozen pretrained Transformer models, but can induce plastic and reconfigurable structures for streaming tasks. We are motivated by some observations of natural intelligence possessed by biological systems (e.g., the human brain) which exhibit remarkable capacity of learning and adapting their structure and function for tackling different tasks throughout their lifespan, while retaining the stability of their core functions. It has been observed in neuroscience that learning and memory are entangled together in a highly sophisticated way (Christophel et al., 2017; Voitov & Mrsic-Flogel, 2022). In this paper, we think, at a high level, of learning model parameters as a process of interacting with the sensory information (data) to convert Short-Term Memory (activations) into a Long-Term Memory (learned parameters), and of selectively adding task-specific parameters as expanding the memory. With this (loose) analogy, we selectively induce learnable parts into Transformers (in contrast to entirely freezing) to induce reconfigurability, selectivity, and plasticity for lifelong learning. We term our framework **ArtiHippo**, which stands for Artificial Hipppocampi, after the hippocampal system that plays an important role in converting Short-Term Memory into Long-Term Memory for lifelong learning.

To this end, we adapt the Learn-to-Grow (L2G) framework (Li et al., 2019) for ViTs (Dosovitskiy et al., 2021), but do not apply Neural Architecture Search (NAS) uniformly across all the layers of a ViT (which is prohibitively computationally expensive). Instead, we aim to find a lightweight component which can preserve and interact with the stable components of a ViT (Dosovitskiy et al., 2021), while inducing plasticity in a lifelong learning setting, i.e., a growing memory. As illustrated in Fig. 2, the final projection layer in the multi-head self-attention (MHSA) block of a ViT is identified and selected as the ArtiHippo (Sec. 3.1). The learn-to-grow NAS is only applied in

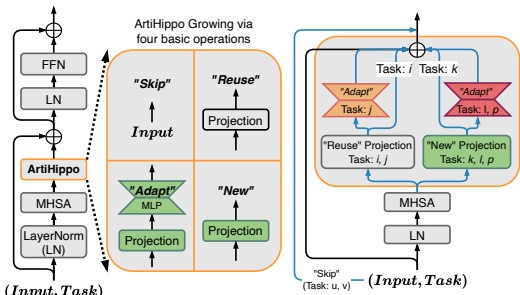

Figure 2: Illustration of the proposed ArtiHippo. *Left:* The Multi-Head Self-Attention (MHSA) block in ViTs (Dosovitskiy et al., 2021) with the proposed Artificial Hippocampi (ArtiHippo) replacing the original linear projection layer. *Middle:* The ArtiHippo growing is maintained by four operations, similar in spirit to the L2G (Li et al., 2019). *Right:* ArtiHippo is represented by a mixture of experts with an example for different tasks (e.g., $j$) starting from $i$.

maintaining ArtiHippo layers, while other components are frozen to maintain the stability of core functions, as illustrated in Fig. 3. We propose a hierarchical task-similarity-oriented exploration-exploitation sampling method (Fig. 4) to account for task synergies and to facilitate a much more effective single-path one-shot (SPOS) NAS (Guo et al., 2020) (Sec. 3.2).

In experiments, this paper considers lifelong learning with task indices available in both training and inference, which is often referred to as *task-incremental setup* (van de Ven et al., 2022). When tasks consist of data from different domains such as the Visual Domain Decathlon (VDD) benchmark (Rebuffi et al., 2017a), it is also related to domain-incremental setup, but without assuming the same output space between tasks. The right of Fig. 2 illustrates an example of learned ArtiHippo. With the task indices, the execution of the computational graph for a given task is straightforward. The proposed method achieves zero-forgetting on old tasks. The proposed method is tested on both the VDD benchmark and the recent 5-Dataset benchmark. It obtains consistently better performance than the prior art with sensible ArtiHippo learned continually. For a comprehensive comparison, we take great efforts in modifying several state-of-the-art methods that were developed for convolutional neural networks (ConvNets) to work with ViTs on VDD.

## 2 RELATED WORK AND OUR CONTRIBUTIONS

*Experience Replay Based approaches* aim to retain some exemplars from the previous tasks and replay them to the model along with the data from the current task (Aljundi et al., 2019b;a; Balaji et al., 2020; Bang et al., 2021; Chaudhry et al., 2021; 2019a; De Lange & Tuytelaars, 2021; Hayes et al., 2020; Lopez-Paz & Ranzato, 2017; Prabhu et al., 2020; Rebuffi et al., 2017b; Chaudhry et al., 2019b; Hayes et al., 2019; Buzzega et al., 2020; Wu et al., 2019; Pham et al., 2021; Cha et al., 2021). Instead of storing raw exemplars, *Generative replay methods* (Shin et al., 2017; Cong et al., 2020) learn the generative process for the data of a task, and replay exemplars sampled from that process along with the data from the current task. For exemplar-free continual learning, *Regularization Based approaches* explicitly control the plasticity of the model by preventing the parameters of the model from deviating too far from their stable values learned on the previous tasks when learning the current task (Aljundi et al., 2018; 2019c; Douillard et al., 2020; Nguyen et al., 2018; Kirkpatrick et al., 2017; Li & Hoiem, 2018; Zenke et al., 2017; Schwarz et al., 2018). Both these approaches aim to balance the stability and plasticity of a fixed-capacity model.

*Dynamic Models* aim to use different parameters per task to avoid use of stored exemplars. Dynamically Expandable Network (Yoon et al., 2018) adds neurons to a network based on learned sparsity constraints and heuristic loss thresholds. PathNet (Fernando et al., 2017) finds task-specific submodules from a dense network, and only trains submodules not used by other tasks. Progressive Neural Networks (Rusu et al., 2016) learn a new network per task and adds lateral connections to the previous tasks' networks. (Rebuffi et al., 2017a) learns residual adapters which are added between the convolutional and batch normalization layers. (Aljundi et al., 2017) learns an expert network per task by transferring the expert network from the most related previous task. The L2G (Li et al., 2019) uses Differentiable Architecture Search (DARTS (Liu et al., 2019a)) to determine if a layer can be reused, adapted, or renewed (3 fundamental skills: reuse, adapt, new) for a task. Our approach is most closely related to Learn to Grow (Li et al., 2019) which can also be interpreted as a Mixture of Experts framework. Veniat et al. (2021) use task priors derived from a task similarity measure and use those to train a stochastic network and retain some layers of the most similar task, and retrain other layers. (Wang et al., 2023) use a task difficulty metric and threshold hyperparameters to either impose regularization constraints on the previous network, to use the same architecture as the

previous tasks, or learn an entirely new architecture and parameters using NAS. Although similar to our method, they rely on (fixed) manually chosen threshold, whereas our method does not have any such heuristics. Dynamic models have also been explored for efficient transfer learning (Morgado & Vasconcelos, 2019; Guo et al., 2019; Mallya et al., 2018).

Recently, there has been increasing interest in lifelong learning using Vision Transformers (Wang et al., 2022d;c; Xue et al., 2022; Ermis et al., 2022; Douillard et al., 2022; Pelosin et al., 2022; Yu et al., 2021; Li et al., 2022; Iscen et al., 2022; Wang et al., 2022a;b; Mohamed et al., 2023; Gao et al., 2023). *Prompt Based approaches* learn external parameters appended to the data tokens that encode task-specific information useful for classification (Wang et al., 2022d;a; Douillard et al., 2022; Smith et al., 2023a). Learning to Prompt (L2P) (Wang et al., 2022d) learns a pool of prompts and uses a key-value based retrieval to retrieve the correct set of prompts at test time. DualPrompt (Wang et al., 2022c) learns generic and task-specific prompts and extends Learning to Prompt. (Xue et al., 2022) uses a ViT pretrained on ImageNet and learns binary masks to enable/disable parameters of the Feedforward Network (FFN), and the attention between image tokens for downstream tasks.

**Our Contributions** We make four main contributions to the field of lifelong learning with ViTs. (i) We propose and identify ArtiHippo in ViTs, i.e., the final projection layers of the multi-head self-attention blocks in a ViT. We also present a new usage for the class-token in ViTs as the memory growing guidance. (ii) We present a hierarchical task-similarity-oriented exploration-exploitation-sampling-based NAS method for learning to grow ArtiHippo continually with respect to four basic growing operations: Skip, Reuse, Adapt, and New to overcome catstrophic forgetting. (iii) We are the first, to the best of our knowledge, to evaluate lifelong learning with ViTs on the large-scale, diverse and imbalanced VDD benchmark (Rebuffi et al., 2017a) with strong empirical performance obtained. We also materialize several state-of-the-art lifelong learning methods that were developed for ConvNets with ViTs on VDD for a comprehensive study. (iv) We show that our method is complementary to prompting-based approaches, and combining the two leads to higher performance.

## 3 APPROACH

In this section, we first present the ablation study on identifying the ArtiHippo in a ViT block (Fig. 2). Then, we present details of learning to grow ArtiHippo (Figs. 3 and 4).

### 3.1 IDENTIFYING ARTIHIPPO IN TRANSFORMERS

The left of Fig. 2 shows a Vision Transformer (ViT) block (Dosovitskiy et al., 2021). Without loss of generality, denote by $x_{L,d}$ an input sequence consisting of $L$ tokens encoded in a $d$-dimensional space. In ViTs, the first token is the so-called class-token. The remaining $L-1$ tokens are formed by patch embedding of an input image, together with additive positional encoding. A ViT block can be expressed as,

$$z_{L,d} = x_{L,d} + \text{Proj}(\text{MHSA}(\text{LN}_1(x_{L,d}))) \quad (1)$$

$$y_{L,d} = z_{L,d} + \text{FFN}(\text{LN}_2(z_{L,d}))) \quad (2)$$

where $\text{Proj}(\cdot)$ is a linear transformation fusing the multi-head outputs from MHSA module.

The MHSA realizes the dot-product self-attention between Query and Key, followed by aggregating with Value, where Query/Key/Value are linear transformatons of the input token sequence. The FFN is often implemented by a multi-layer perceptron (MLP) with a feature expansion layer $\text{MLP}^u$ and a feature reduction layer $\text{MLP}^d$ with a nonlinear activation function (such as GELU) in the between.

| Index | Finetuned Component | Avg. Acc. | Avg. Forgetting |
|---|---|---|---|
| 1 | $\text{LN}_1 + \text{LN}_2$ | 81.76 | 21.24 |
| 2 | FFN | 84.20 | 44.76 |
| 3 | $\text{MLP}^d$ | 83.66 | 37.99 |
| 4 | $\text{LN}_2$ | 80.04 | 16.35 |
| 5 | $\text{MHSA} + \text{LN}_1$ | 85.26 | 54.38 |
| 6 | $\text{LN}_1$ | 81.18 | 19.04 |
| 7 | Query | 81.57 | 19.69 |
| 8 | Key | 81.56 | 19.19 |
| 9 | Query+Key | 81.49 | 31.10 |
| 10 | Value | 84.99 | 37.58 |
| 11 | Projection (**ArtiHippo**) | 85.11 | 30.50 |
| | Classifier w/ Frozen Backbone | 70.78 | - |

Table 1: Ablation study on identifying the Arti-Hippo in a Transformer block (Eqns. 1 and 2). See text for detail. The last row shows the result of a conventional transfer learning setting in which only the head classifier is trained.

For resilient task-incremental lifelong learning using ViTs, **our very first step is to investigate whether there is a simple yet expressive "sweet spot" in the Transformer block that plays the functional role of Hippocampi in the human brain (i.e., ArtiHippo)**, that is converting short-term streaming task memory into long-term memory to support lifelong learning without catastrophic forgetting. The proposed identification process is straightforward. Without introducing any modules handling forgetting, we compare both the task-to-task forward transferrability and the sequential

forgetting of different components in a Transformer block. Our intuition is that a desirable Arti-Hippo component must enable strong transferrability with manageable forgetting. Table 1 shows the abaltion study of identifying "sweet spot" candidates. **Due to space limit, please refer to the Appendix B** for the detailed experimental settings, the definitions of average accuracy (Eqn. 3) and average forgetting (Eqn. 4), and our analyses.

In sum, due to the strong forward transfer ability, manageable forgetting, maintaining simplicity and for less invasive implementation in practice, **we select the Projection layer in the MHSA block as ArtiHippo to develop our proposed long-term task-similarity-oriented memory based lifelong learning.** Through ablation studies in Appendix C.2 Table 6, we verify that using the Projection layer is better than the Value layer. We also show that using other layers of the ViT leads to significantly worse performance while using the FFN obtains similar performance as that of the Projection layer in line with Table 1 but at a much higher parameter cost, thus empirically validating our hypothesis.

### 3.2 LEARNING TO GROW ARTIHIPPO CONTINUALLY

This section presents details of learning to grow ArtiHippo based on NAS. As illustrated in Fig. 3, for a new task $t$ given the network learned for the first $t-1$ tasks, it consists of three components: the Supernet construction (the parameter space of growing Arti-Hippo), the Supernet training (the parameter estimation of growing ArtiHippo), and the target network selection and finetuning (the consolidation of the ArtiHippo for the task $t$).

#### 3.2.1 SUPERNET CONSTRUCTION

We start with a vanilla $D$-layer ViT model (e.g., the 12-layer ViT-Base) (Dosovitskiy et al., 2021) and train it on the first task (e.g., Ima-geNet in the VDD benchmark (Rebuffi et al., 2017a)) following the conventional recipe. The proposed ArtiHippo is represented by a mix-ture of experts, similar in spirit to (Ruiz et al., 2021). After the first task, the ArtiHippo at the $l$-th layer in the ViT model consists of a sin-gle expert which is defined by a tuple, $E^{l,1} = (P^{l,1}, \mu^{l,1})$, where $P^{l,1}$ is the projection layer and $\mu^{l,1} \in R^d$ is the associated mean class-token pooled from the training dataset after the model is trained. Without loss of generality, we consider how the growing space of ArtiHippo is constructed at a single layer, assuming the cur-

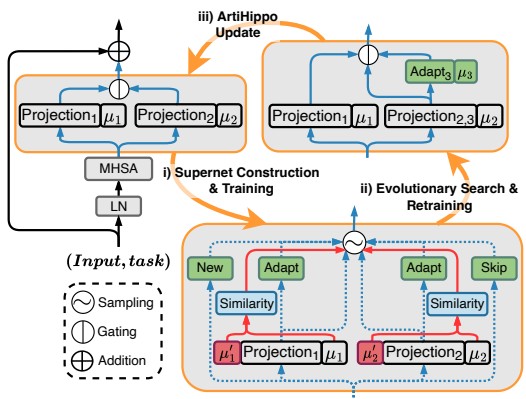

Figure 3: Illustration of ArtiHippo growing via NAS using the four learning-to-grow operations in lifelong learning. See text for details.

rent ArtiHippo consists of two experts, $\{E^{l,1}, E^{l,2}\}$ (Fig. 3, left). We utilize four operations in the Supernet construction:

- `skip`: Skips the entire MHSA block (i.e., the hard version of the drop-path method widely used in training ViT models), which encourages the adaptivity accounting for the diverse nature of tasks.
- `reuse`: Uses the projection layer from an old task for the new task unchanged (including associ-ated mean class-token), which will help task synergies in learning.
- `adapt`: Introduces a new lightweight layer on top of the projection layer of an old task, imple-mented by a MLP with one squeezing hidden layer, and a new mean class-token computed at the added adapt MLP layer. We propose a hybrid adapter which acts as a plain 2-layer MLP during searcch, and a residual MLP during finetuning (details in the Appendix Sec. 2.1).
- `new`: Adds a new projection layer and a mean class-token, enabling the model to handle corner cases and novel situations.

The bottom of Fig. 3 shows the growing space. The Supernet is constructed by `reusing` and `adapting` each existing expert at layer $l$, and adding a `new` and a `skip` expert. The newly added `adapt` MLPs and projection layers will be trained from scratch using the data of a new task only.

#### 3.2.2 SUPERNET TRAINING

To train the Supernet constructed for a new task $t$, we build on the SPOS method (Guo et al., 2020) due to its efficiency. The basic idea of SPOS is to sample a single-path sub-network from the Super-

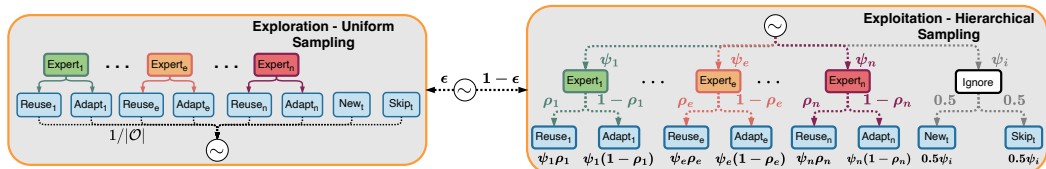

Figure 4: Illustration of the proposed exploration-exploitation sampling based SPOS NAS (Guo et al., 2020) for lifelong learning. It harnesses the best of the vanilla pure exploration strategy (top) and the proposed exploitation strategy (bottom) using a simple epoch-wise scheduling. See text for details.

net by sampling an expert at every layer in each iteration (mini-batch) of training. One key aspect is the sampling strategy. The vanilla SPOS method uses uniform sampling (i.e., the *pure exploration* strategy, Fig. 4 left), which has the potential of traversing all possible realizations of the mixture of experts of the ArtiHippo in the long run, but may not be desirable (or sufficiently effective) in a lifelong learning setup because it ignores inter-task similarities/synergies. To overcome this, we propose an exploitation strategy (Fig. 4 right), which utilizes a hierarchical sampling method that forms the categorical distribution over the operations in the search space explicitly based on task similarities. We first present details on the task-similarity oriented sampling in this section. Then we present an exploration-exploitation integration strategy to harness the best of the two in Sec. 3.2.4.

**Task-Similarity Oriented Sampling**: Let $\mathbb{E}^l$ be the set of Experts at the $l$-the layer learned till task $t - 1$. For each candidate expert $e \in \mathbb{E}^l$, we first compute the mean class-token for the $t$-task, $\hat{\mu}_e^t$ using the current model, and compute the task similarity between the $t$-th task and the Expert $e$ as $S_e(t) = \text{NormCosine}(\hat{\mu}_e^t, \mu_e)$, where $\text{NormCosine}(\cdot, \cdot)$ is the Normalized Cosine Similarity, which is calculated by scaling the Cosine Similarity score between $-1$ and $1$ using the minimum and the maximum Cosine Similarity scores from all the experts in all the MHSA blocks of the ViT. This normalization is necessary to increase the difference in magnitudes of the similarities between tasks, which results in better Expert sampling distributions during the sampling process in our experiments. The similarities for the newly added new and skip experts cannot be calculated, since the parameters are randomly initialized. Instead, we treat them together as a pseudo *Ignore* expert, for which the similarity is calculated as $S_{ig}(t) = -\max_e S_e(t)$. Intuitively, this means we only ignore the previous experts in proportion to the best possible expert (which has the maximum similarity).

The categorical distribution is then computed via Softmax across all the scores of all the Experts at a layer. The probability of sampling a candidate Expert $e \in \mathbb{E}^l \cup \{\mathbb{E}^l_{ignore}\}$ is defined by $\psi_e = \frac{\exp(S_e(t))}{\sum_{e' \in \mathbb{E}^l \cup \{\mathbb{E}^l_{ignore}\}} \exp(S_{e'}(t))}$. With an Expert $e$ sampled (with a probability $\psi_e$), we further compute its retention Bernoulli probability via a Sigmoid transformation of the task similarity score defined by $\rho_e = \frac{1}{1 + \exp(-S_e(t))}$. If the sampled Expert $e \in \mathbb{E}^l$, we sample the Reuse operation with a probability $\rho_e$ and the Adapt operation with probability $1 - \rho_e$. If the Expert $e = \mathbb{E}^l_{ignore}$, we randomly sample from the two operations: Skip and New with a probability 0.5.

### 3.2.3 TARGET NETWORK SELECTION AND FINETUNING

After the Supernet is trained, we adopt the same evolutionary search used in the SPOS method (Guo et al., 2020) based on the proposed hierarchical sampling strategy. The evolutionary search is performed on the validation set to select the path which gives the best validation accuracy. After the target network for a new task is selected, we retrain the newly added layers by the New and Adapt operations from scratch (random initialization), rather than keeping or warming-up from the weights from the Supernet training. This is based on the observations in network pruning that it is the neural architecture topology that matters and that the warm-up weights may not need to be preserved to ensure good performance on the target dataset (Liu et al., 2019b).

### 3.2.4 BALANCING EXPLORATION AND EXPLOITATION:

As illustrated in Fig. 4, to harness the best of the pure exploration strategy and the proposed exploitation strategy, we apply epoch-wise exploration and exploitation sampling for simplicity. At the beginning of an epoch in the Supernet training, we choose the pure exploration strategy with a probability of $\epsilon_1$ (e.g., 0.3), and the hierarchical sampling strategy with a probability of $1 - \epsilon_1$. Similarly, when generating the initial population during the evolutionary search, we draw a candidate target network from a uniform distribution over the operations with a probability of $\epsilon_2$, and

| Method | ImNet | C100 | SVHN | UCF | OGlt | GTSR | DPed | Flwr | Airc. | DTD | Avg. Accuracy |
|---|---|---|---|---|---|---|---|---|---|---|---|
| S-Prompts† (L=12) (Wang et al., 2022a) | 82.65 | 89.32 | 88.91 | 64.52 | 72.17 | 99.29 | **99.89** | **96.93** | 45.55 | **60.76** | 80.00 ± 0.07 |
| L2P† (L=12) (Wang et al., 2022d) | 82.65 | 89.32 | 89.89 | 65.63 | 72.34 | 99.55 | **99.94** | 96.63 | 45.24 | 59.57 | 80.08 ± 0.10 |
| L2G-ResNet26 (DARTS) (Li et al., 2019) | 69.84 | 79.59 | 95.28 | 72.03 | **86.6** | 99.72 | 99.52 | 71.27 | **53.01** | 49.89 | 77.68 |
| *L2G (Li et al., 2019) (DARTS) | 82.65 | 88.47 | 85.20 | **79.22** | 80.19 | 99.28 | 98.06 | 76.14 | 39.29 | 46.01 | 77.45 ± 2.41 |
| *L2G (Li et al., 2019) (β-DARTS) | 82.65 | 88.95 | 94.73 | 75.31 | 79.76 | 99.84 | 99.76 | 78.86 | 34.50 | 47.09 | 78.14 ± 0.54 |
| Our ArtiHippo (Uniform, 150 epochs) | 82.65 | 76.20 | 95.60 | 75.14 | 80.72 | **99.92** | 99.86 | 76.41 | 42.74 | 41.74 | 77.10 ± 0.75 |
| Our ArtiHippo (Hierarchical, 50 epochs) | 82.65 | **90.97** | 96.05 | 75.20 | 82.36 | **99.91** | 99.58 | 87.16 | 42.10 | 52.54 | **80.85 ± 0.72** |
| Our ArtiHippo (Hierarchical, 150 epochs) | 82.65 | **90.86** | **96.06** | 75.63 | 84.06 | **99.92** | 99.83 | 89.28 | **51.94** | 55.78 | **82.60 ± 0.55** |
| Our ArtiHippo (Hierarchical+L=1, 150 epochs) | 82.65 | 90.50 | **96.19** | **79.70** | **85.71** | 99.91 | 99.83 | 92.42 | **52.23** | 58.99 | **83.55 ± 0.09** |

Table 2: Results on the VDD benchmark (Rebuffi et al., 2017a). Our ArtiHippo shows clear improvements over the previous approaches. All the results from our experiments are averaged over 3 different seeds. The 2 highest accuracies per task have been highlighted. All the methods use the same ViT-B/8 backbone containing 86.04M parameters and having 14.21G FLOPs unless otherwise stated. † our modifications for the task-incremental setting. * our reproduction with the vanilla L2G method (Li et al., 2019) for the ViT backbone. "Hierarchical+$L = 1$" means the integration between our ArtiHippo and the L2P method (Sec. 3.3).

from the hierarchical sampling process with a probability of $1 - \epsilon_2$, respectively. In practice, we set $\epsilon_2 > \epsilon_1$ (e.g., $\epsilon_2 = 0.5$) to encourage more exploration during the evolutionary search, while encouraging more exploitation for faster learning in the Supernet training. Our experiments show that this exploration-exploitation strategy achieves higher Average Accuracy and results in a lower parameter increase than pure exploration by a large margin (Fig. 6 in the Appendix).

### 3.3 INTEGRATING ARTIHIPPO WITH LEARNING TO PROMPT

Since prompting-based methods (Wang et al., 2022d;c;a; Douillard et al., 2022) are complimentary to our proposed method, we propose a simple method for harnessing the best of both, and show that this leads to further improvement. At the beginning of the Supernet training, a task-specific classification token is learned using the ImageNet backbone (similar to S-Prompts (Wang et al., 2022a)). Then, instead of using the `cls` token from the ImageNet task, we used the learned task token during NAS. When finetuning the learned architecture, we first train the task token using the fixed ImageNet backbone, and then use this trained token to train the architecture components.

## 4 EXPERIMENTS

In this section, we test the proposed method on two benchmarks and compare with the prior art. We evaluate our method in the task-incremental setting, where each task contains a disjoint set of classes and/or domains and task index is available during inference. The proposed method obtains better performance than the prior art in comparisons. **Our PyTorch source code is provided with the supplementary material.** Due to space limitations, we provide the implementation details in the Appendix. We use 1 Nvidia A100 GPU for experiments on the VDD and 5-Dataset benchmarks.

**Data and Metrics**: We evaluate our approach on the Visual Domain Decathlon (VDD) datasets (Rebuffi et al., 2017a) and the 5-Datasets benchmark introduced in (Ebrahimi et al., 2020). Each of the individual dataset in these two benchmarks are treated as separate non-overlapping tasks. The VDD benchmark is challenging because of the large variations in tasks as well as small number of samples in many tasks, which makes it a favorable for evaluating lifelong learning algorithms. *Details of the benchmarks are provided in the Appendix H*. Since catastrophic forgetting is fully addressed by our method, we evaluate the performance of our method using the average accuracy defined by Eqn. 5 in the Appendix B.

**Baselines:** We compare with Learn to Grow (L2G) (Li et al., 2019), Learning to Prompt (L2P) (Wang et al., 2022d) and S-Prompts (Wang et al., 2022a). We use our implementation for evaluating L2P and S-Prompts, and take efforts of modifing them to work in the task-incremental setting for fair comparisons, denoted as L2P† and S-Prompts† in Table 2. We compare L2G with DARTS, and with more advanced β-DARTS (Ye et al., 2022). We provide the modification and implementation details, the choice for the number of prompts per task ($L$) in L2P, and ablations for the number of prompts in the Appendix K. For completeness, we also compare with a baseline of Experience Replay (Rebuffi et al., 2017b), L2 Parameter Regularization (Smith et al., 2023b), and Elastic Weight Consolidation (Kirkpatrick et al., 2017), shown in Tab. 7 in the Appendix due to space limit. The three methods suffer from catastrophic forgetting significantly.

## 4.1 Results and Analysis on the VDD Benchmark

Table 2 shows the results and comparisons. Our method shows consistent performance improvement across tasks compared to Learn to Grow, S-Prompts[†] and L2P[†]. Following is an analysis of the results showing the advantages and effectiveness of our proposed method for ViTs:

- Table 2 shows that L2P[†] (which uses ViTB/8) performs better than L2G (which uses ResNet26), showing that prompting based methods which leverage the robust features learned by the ViT are indeed effective for lifelong learning.

- However, a closer look at task level accuracies shows that for tasks which are significantly different from the base task of the ViT (Omniglot, UCF101, SVHN), L2G significantly outperforms L2P[†], showing that introducing new parameters can be beneficial, which justifies our motivation for for seeking more integrative memory mechanisms.

- The poor average accuracy of L2G, when applied to the attention projection layer of ViTs shows that L2G is ill suited for learning to grow for ViTs, and even Uniform sampling (original SPOS) can obtain similar average accuracy. Our proposed Hiererchical sampling method outperforms both, prompting-based methods and L2G.

- Our hierarchical sampling scheme performs better than pure exploration (which uses 150 epochs of supernet training) with just 50 epochs of supernet training, which shows the effectiveness of the proposed sampling strategy. In fact, pure exploration cannot match the hierarchical sampling even if the supernet is trained for 300 epochs, as shown in Fig. 6 in the Appendix.

- S-Prompts[†] and L2P[†] perform better for tasks which are similar to the base task and have very less data (VGG-Flowers, DTD), which suggests that prompting based methods are more data-efficient. Thus, prompt-based methods and our growing-based method are complementary and combining them leads to even better performance (Tab. 2). Section 3.3 describes a preliminary approach to combine the two approaches. A more comprehensive integration of ArtiHippo with prompt based approaches is left for future work.

## 4.2 Results on the 5-Dataset Benchmark

Table 3 shows the comparisons. We use the same ViT-B/8 backbone pretrained on the ImageNet images from the VDD benchmark for all the experiments across all the methods and upsample the images in the 5-Dataset benchmark (consisting of CIFAR10 (Krizhevsky et al., 2009), MNIST (Lecun et al., 1998), Fashion-MNIST (Xiao et al., 2017), not-MNIST (Bulatov, 2011), and SVHN (Netzer et al., 2011)) to $72 \times 72$. We can see that ArtiHippo sig-

| Method | #Prompts | Avg. Acc. |
|---|---|---|
| S-Prompts[†] | 1 | $88.93 \pm 0.34$ |
| S-Prompts[†] | 12 | $92.42 \pm 0.11$ |
| L2P[†] | 12 | $92.73 \pm 0.10$ |
| L2G (DARTS) | - | $93.88 \pm 2.86$ |
| L2G ($\beta$-DARTS) | - | $92.19 \pm 1.48$ |
| ArtiHippo (Projection) | - | $\mathbf{94.33 \pm 0.45}$ |

Table 3: Results on the 5-Dataset benchmark. The results have been averaged over 5 different task orders.

nificantly outperforms Learn to Grow, L2P[†], S-Prompts[†] under the task-incremental setting.

## 4.3 Learned architecture and architecture efficiency

Fig. 5 shows the experts learned and the long-term memory structures formed at each block tasks in the VDD dataset. When learning CIFAR100 after ImageNet, the search process learns to reuse most of the ImageNet experts. This is an intuitive result since both the tasks represent natural images. Task-specific structures can be observed between related tasks. For example, at B1, the search process learns to adapt the ImageNet expert when lerning SVHN, which is further adapted when learning Omniglot. At B3, the search process learns to adapt ImageNet when learning SVHN, which is then reused for Omniglot. At B12, the search process learns to adapt the new expert learned for CIFAR100 for SVHN, which is subsequently adapted again for Omniglot and GTSRB (three tasks which rely on some form of symbol recognition). At Block B3, the expert learned for UCF101 is reused for Pedestrian Classification (UCF101 is an action recognition benchmark, making it similar to DPed). Some blocks have more synergy: the search process reuses the ImageNet experts at blocks B2 and B11 for all the downstream tasks. The sensible architectures learned continually show the effectiveness of the proposed task-similarity-oriented ArtiHippo.

In addition to learning qualitatively meaningful architectures, the proposed method also shows quantitative advantages. On the VDD dataset, the number of parameters increases by 0.68M/task (averaged over 3 different runs). Although this is higher than L2P, our method increases the number of FLOPs only by 0.23G/task (averaged over 3 different runs), which is advantageous as compared to the increase of 2.02G/task of the L2P.

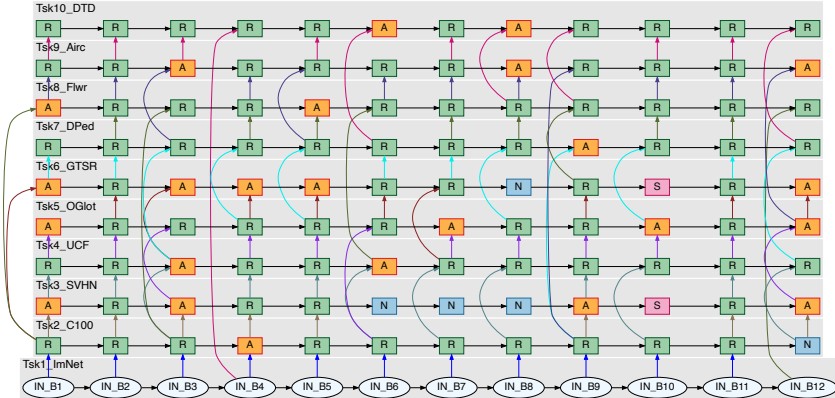

Figure 5: ArtiHippo learned-to-grow on VDD. Starting from the ImageNet-pretrained ViT (B1 – B12 in Tsk1_ImNet), sensible architectures are continually learned for the remaining 9 tasks on the VDD dataset. Each column denotes a Transformer block of the ViT in which only the projection layer of the MHSA block is maintained as ArtiHippo with the remaining components frozen. S, R, A, N represent Skip, Reuse, Adapt and New respectively. *Best viewed in color and magnification.*

### 4.4 CAN ARTIHIPPO EFFICIENTLY LEVERAGE A STRONG BACKBONE?

| Method-Backbone | ImNet | C100 | SVHN | UCF | OGlt | GTSR | DPed | Flwr | Airc. | DTD | Avg. Accuracy |
|---|---|---|---|---|---|---|---|---|---|---|---|
| SupSup-Proj (Wortsman et al., 2020) | 82.65 | 89.96 | **96.05** | **81.68** | **84.60** | **99.97** | **99.97** | 78.76 | 44.18 | 51.60 | 81.14 ± 0.04 |
| EFT (Verma et al., 2021) | 82.65 | **91.86** | 93.51 | 73.89 | 75.62 | 99.58 | **99.98** | **96.34** | 48.17 | **64.40** | 82.60 ± 0.07 |
| Lightweight Learner (Ge et al., 2023b) | 82.65 | **91.92** | 93.90 | 75.63 | 77.07 | 99.71 | 99.96 | **96.47** | 49.33 | 64.34 | **83.10 ± 0.02** |
| ArtiHippo-T2T (50 epochs) | 82.65 | 90.93 | 95.96 | 80.74 | 83.25 | 99.94 | 99.96 | 94.12 | **58.90** | 60.05 | **84.65 ± 0.33** |

Table 4: Results on the VDD benchmark (Rebuffi et al., 2017a) under the task-to-task based lifelong learning setting. Our ArtiHippo-T2T shows clear improvements over the previous approaches. The 2 highest accuracies per task have been highlighted. All the methods use the same ViT-B/8 backbone.

Recently, there has been increasing interest in learning techniques which can effectively leverage a pretrained backbone, and add task-specific parameters for lifelong learning, which we call as a task-to-task (T2T) setting. We evaluate our method in this setting by modifying ArtiHippo to always learn from the ImageNet trained ViT model and evaluate on the VDD benchmark. We compare with recent methods which propose a similar setting: Supermasks in Superposition (SupSup, (Wortsman et al., 2020)), Efficient Feature Transformation (EFT, (Verma et al., 2021)), and Lightweight Learner (LL, (Ge et al., 2023b)). We take efforts to modify all the methods to work for ViTs, and provide the details in Appendix F. As seen in Table 4 ArtiHippo-T2T outperforms all the other methods with just 50 epochs of supernet training, making it very efficient. Table 4 shows that SupSup achieves a higher accuracy on tasks which are very different from the original task on which the backbone is trained (Omniglot and UCF101) but does not perform well on tasks which are similar to the base task (VGG-Flowers, Aircraft, DTD), whereas EFT and LL shows the exact opposite behavior. Our ArtiHippo-T2T can perform well across all the tasks and also achieves the highest average accuracy, which shows that our method is more generic across tasks of different nature.

## 5 DISCUSSION AND CONCLUSION

This paper presents a method of transforming Vision Transformers (ViTs) for resilient lifelong learning under the task-incremental setting. It learns to dynamically grow the final projection layer of the multi-head self-attention of a ViT in a task-aware way using four operations, Skip, Reuse, Adapt and New. The final projection layer is identified as the Artificial Hippocampi (ArtiHippo) of ViTs. The learning-to-grow of ArtiHippo is realized by our proposed hierarchical exploration-exploitation sampling based single-path one-short Neural Architectural Search (NAS), where the exploitation utilizes task similarities (synergies) defined by the normalized cosine similarity between the mean class tokens of a new task and those of old tasks. In experiments, the proposed method is tested on the challenging VDD and the 5-Datasets benchmarks. We also take great efforts in materializing several state-of-the-art baseline methods for ViTs and tested on the VDD. It obtains better performance than the prior art with sensible ArtiHippo learned continually.

Our future work is to adapt the proposed ArtiHippo for handling class-incremental lifelong learning, and hopefully for fine-tuning large language models, or large foundation models in general.

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

## A  APPENDIX

In the Appendix, we elaborate on the following aspects that are not presented in the submission due to space limit:

- **Source Code and Training Logs:** The code for our implementation on the VDD and 5-datasets benchmarks is available in the directory `artihippo`. Due to size limit, we are not able to upload the pretrained checkpoints. We provide the anonymized training logs of the experiments on the VDD benchmark and the 5-datasets benchmarks in `artihippo/artifacts`. The pretrained checkpoints will be released after the reviewing process.

- **Section B: Identifying ArtiHippo in Transformers**: We provide the analysis which leads us to identify the final linear projection layer from the Multi-Head Self-Attention block as the Arti-Hippo.

- **Section C: Ablation Studies**:

    - **Section C.1**: We describe the proposed hybrid adapter and an ablation study to verify its effect.
    - **Section C.2**: We perform ablation experiments on the feasibility of other components of the Vision Transformer, and verify our hypothesis of selecting the final linear projection layer of the MHSA block as the ArtiHippo (Section 3.1 in the main text).
    - **Section C.3**: Through ablation experiments, we show that the proposed exploration-exploitaiton sampling strategy obtains higher average accuracy and model efficiency, and requires less training epochs for the supernet.
    - **Section C.4**: We show that our proposed exploration-exploitation sampling strategy can limit the parameter growth over time.

- **Section D: Comparison with Additional Methods**: We compare with Elastic Weight Consolidation (EWC, Kirkpatrick et al. (2017)), L2 Parameter Regularization (Smith et al., 2023b), and Experience Replay (Rebuffi et al., 2017b) for completeness.

- **Section E: Discussion on other Transformer-based methods**: We discuss other lifelong learning methods that use Vision Transformers, and the similarities and differences with them.

- **Section F: Modifying dynamic model based methods for vision transformers**: We provide details of our modifications to Supermasks in Superposition (SupSup, (Wortsman et al., 2020)), Efficient Feature Transformations (EFT, (Verma et al., 2021)), and Lightweight Learner (LL, (Ge et al., 2023b)) for Vision Transformers.

- **Section G: Implementation details for S-Prompts and Learn to Prompt**: We describe our implementation of L2P[†] and S-Prompts[†] used for comparisons, show the results of ablation studies for the number of prompts used in S-Prompts (Wang et al., 2022a), and describe how we calculate the number of prompts used in L2P[†].

- **Section H: Details of the two benchmarks**: the Visual Domain Decathlon (VDD) (Rebuffi et al., 2017a) benchmark (Section H.1) and the 5-Datasets (Ebrahimi et al., 2020) benchmark (Section H.2).

- **Section I: The Base Model and Its Training Details**: the Vision Transformer (ViT) model specification (ViT-B/8) used in our experiments on the VDD and 5-Dataset benchmarks, and training details on the ImageNet (Section I.1).

- **Section J: Background:** To be self-contained, we give a brief introduction to the learn-to-grow method (Li et al., 2019) in Section J.1 and the single-path one-shot (SPOS) neural architecture search (NAS) algorithm (Guo et al., 2020) in Section J.2.

- **Section K: Settings and Hyperparameters in the Proposed Lifelong Learning:** We provide the hyperparameters used for training on the VDD and 5-dataset benchmarks.

- **Section L: Learned architecture with pure exploration and different task order**: We show that with a different task order, the proposed method can still learn to exploit inter-task similarities. We also show that pure exploration cannot effectively exploit task similairties by comparing the learned architecture with the architecture learned using the exploration-exploitation strategy.

# B  IDENTIFYING ARTIHIPPO IN TRANSFORMERS

We provide the definitions of accuracy and forgetting used in Section 3.1 and analysis of how we choose the final linear Projection layer as the ArtiHippo. We first compare the transferability of the ViT trained with the first task, ImageNet to the remaining 9 tasks in a pairwise task-to-task manner and compute the average Top-1 accuracy on the 9 tasks from the VDD benchmark (Rebuffi et al., 2017a). Then, we start with the ImageNet-trained ViT, and train it on the remaining 9 tasks continually and sequentially in a predefined order with the average forgetting (Chaudhry et al., 2018) on the first 9 tasks (including ImageNet) compared. As shown in Table 1, we compare 11 components or composite components across all blocks in the ImageNet-pretrained ViT.

Let $T_1, T_2, \cdots, T_N$ be a sequence of $N$ tasks (e.g., $N = 10$ in the VDD benchmark). A model consists of a feature backbone and a task head classifier. Let $f_{T_{n|1}}$ be the backbone trained for the task $n$ ($n = 2, ..N$) with weights initialized from the model of task 1, and the learned head classifier $C_n$ from scratch. The average transfer learning accuracy of the first task model to the remaining $N - 1$ tasks is defined as:

$$A_N = \frac{1}{N-1} \sum_{n=2}^{N} \text{Acc}(T_n; f_{T_{n|1}, C_n}) \tag{3}$$

where Accuracy() uses the Top-1 accuracy in classification.

Let $f_{T_{1:n}}$ be the backbone trained sequentially after task $T_n$ and and $C_n$ the head classifier trained for task $T_n$. Denote by $a_{n,i} = \text{Accuracy}(T_i; f_{T_{1:n}}, C_i)$, the accuracy on the task $i$ using the backbone that has been trained on tasks from 1 to $n$ ($i < n$). The average forgetting on the first $N - 1$ tasks is:

$$\mathbb{F}_N = \frac{1}{N-1} \sum_{n=1}^{N-1} \left( \max_{j \in [n, N-1]} a_{j,n} - a_{N,n} \right) \tag{4}$$

The average accuracy in lifelong learning is defined by,

$$\mathbf{A}_N = \frac{1}{N} \sum_{i=n}^{N} a_{N,n}, \tag{5}$$

where $N$ is the total number of tasks, and $a_{n,i} = \text{Accuracy}(T_i; f_{T_{1:n}}, C_i)$

From Table 1, **the following observations lead us to select the projection layer as ArtiHippo**:

(i) Continually finetuning the entire MHSA block (i.e., MHSA+LN$_1$) obtains the best average accuracy, which has been observed in (Touvron et al., 2022) in terms of finetuning ImageNet-pretained ViTs on downstream tasks. However, (Touvron et al., 2022) does not consider lifelong learning settings, and as shown here finetuning the entire MHSA block incurs the highest average forgetting, which means that it is task specific. Continually finetuning the entire FFN block (i.e., MLP$^{\text{down}}$+MLP$^{\text{up}}$+LN$_2$) has a similar effect as finetuning the entire MHSA block. In the literature, the Vision Mixture of Expert framework (Ruiz et al., 2021) where an expert is formed by an entire MLP block takes advantage of the high average performance preservation.

(ii) In lifelong learning scenarios, maintaining either the entire MHSA block or the entire FFN block could address the catastrophic forgetting, but at the expense of high model complexity and heavy computational cost in both learning and inference.

(iii) The final projection layer and the Value layer in the MHSA block, which have been overlooked, can maintain high average accuracy (as well as manageable average forgetting, to be elaborated). It is also much more "affordable" to maintain it in lifelong learning, especially with respect to the four basic growing operations (`skip`, `reuse`, `adapt` and `new`). Intuitively, the final projection layer is used to fuse multi-head outputs from the self-attention module. In ViTs, the self-attention module is used to mix/fuse tokens spatially and it has been observed in MetaFormers (Yu et al., 2022a;b) that simple local average pooling and even random mixing can perform well. So, it makes sense to keep the self-attention module frozen from the first task and maintain the projection layer to fuse the outputs. However, the Value layer is implemented as a parallel computation along with the Key and Query, which makes it inefficient to incorporate into the Mixture of Experts framework.

Through ablation studies (Appendix C.2), we show that using the projection layer as ArtiHippo achieves higher performance than Query and Key, while achieving almost the same performance as that of the FFN but with much smaller number of parameters.

| | | | | | | Learned Operation per Block | | | | | | | | | | | |
|---|---|---|---|---|---|---|---|---|---|---|---|---|---|---|---|---|---|
| | `Adapter` in | | #Param Added | Rel. ↑ | Test Acc. | 1 | 2 | 3 | 4 | 5 | 6 | 7 | 8 | 9 | 10 | 11 | 12 |
| | NAS | Finetune | | | | | | | | | | | | | | | |
| Shorcut in | w/o A & S | w/ A | 2.96M | 3.47% | 82.18 | A | A | R | R | A | R | A | N | N | N | S | S |
| `Adapter` | | w/o S | | | 78.16 | | | | | | | | | | | | |
| | w/ S & A | w/ S & A | 4.14M | 4.89% | 82.32 | A | A | A | A | A | A | N | A | A | N | A | N |

Table 5: Results of the ablation study on the `Adapter` implementation (Section 3.2.1): with (w/) vs without (w/o) shortcut connection for the MLP `Adapt` layer. We test lifelong learning from ImageNet to Omniglot in the VDD. The proposed combination of w/o shortcut in Supernet NAS training and target network selection and w/ shortcut in finetuning (retraining newly added layers) is the best in terms of the trade-off between performance and cost.

## C   ABLATION STUDIES

### C.1   THE STRUCTURE OF `Adapter`

**How to `Adapt` in a sustainable way?**   The proposed `Adapt` operation will effectively increase the depth of the network in a plain way. In the worst case, if too many tasks use `Adapt` on top of each other, we will end up stacking too many MLP layers together. This may lead to unstable training due to gradient vanishing and exploding. Shortcut connections (He et al., 2016) have been shown to alleviate the gradient vanishing and exploding problems, making it possible to train deeper networks. Due to this residual architecture, the training can ignore an adapter if needed, and leads to a better performance. However, in the lifelong learning setup, where subsequent tasks might have different distributions, the search process might disproportionately encourage `Adapt` operations because of this ability. To counter this, we propose a hybrid `Adapter` which acts as a plain 2-layer MLP during Supernet training and target network selection, and a residual MLP during finetuning. With an ablation study (Table 1 in Supplementary), we show that much more compact models can be learned with negligible loss in accuracy.

We verify the effectiveness of the proposed hybrid adapter using a lifelong learning setup with 2 tasks: ImageNet and Omniglot. The Omniglot dataset presents two major challenges for a lifelong learning system. First, Omniglot is a few-shot dataset, for which we may expect a lifelong learning system can learn a model less complex than the one for ImageNet. Second, Omniglot has a significantly different data distribution than ImageNet, for which we may expect a lifelong learning system will need to introduce new parameters, but hopefully in a sensible and explainable way. Table 5 shows the results. In terms of the learned neural architecture, a more compact model (row 3) is learned without the shortcut in the adapter during Supernet training and target network selection: the last two MHSA blocks are skipped and three blocks are reused. Skipping the last two MHSA blocks makes intuitive sense since Omniglot may not need those high-level self-attention (learned for ImageNet) due to the nature of the dataset. The three consecutive `new` operations (in Blocks 8,9,10) also make sense in terms of learning new self-attention fusion layers (i.e., new memory) to account for the change of the nature of the task. Adding shortcut connection back in the finetuning shows significant performance improvement (from 78.16% to 82.18%), making it very close to the performance (82.32%) obtained by the much more expensive and less intuitively meaningful alternative (the last row).

| Component | ImNet | C100 | SVHN | UCF | OGlt | GTSR | DPed | Flwr | Airc. | DTD | Avg. Accuracy | Avg. Param. Inc./task (M) |
|---|---|---|---|---|---|---|---|---|---|---|---|---|
| Projection | 82.65 | **90.86** | **96.06** | 75.63 | 84.06 | **99.92** | 99.83 | **89.28** | **51.94** | 55.78 | 82.60 ± 0.55 | 1.12 ± 0.03 |
| Value | 82.65 | 85.59 | 95.82 | 72.25 | **84.20** | 99.89 | 99.89 | 84.05 | 45.80 | 53.37 | 80.35 ± 1.10 | 1.73 ± 0.13 |
| Query | 82.65 | 90.00 | 94.56 | 70.08 | 78.56 | 99.83 | 99.91 | 85.00 | 43.37 | 55.53 | 79.95 ± 0.76 | 2.65 ± 0.16 |
| Key | 82.65 | 89.57 | 94.41 | 71.31 | 81.12 | 99.89 | **99.92** | 87.03 | 45.10 | **56.01** | 80.00 ± 0.70 | 2.47 ± 0.23 |
| FFN | 82.65 | 90.70 | **96.18** | 79.59 | 85.44 | **99.92** | **99.92** | 87.19 | 51.66 | 54.02 | **82.73 ± 1.06** | 4.44 ± 0.83 |

Table 6: Results of ablation study on the other components of the ViT used for realizing the Arti-Hippo. Realizing ArtiHippo at the FFN shows slightly better performance than the Projection layer. However, the Projection layer is much more parameter efficient than the FFN. Using the Projection layer offers negligible drop in Average Accuracy without sacrificing parameter efficiency. The results have been averaged over 3 different seeds.

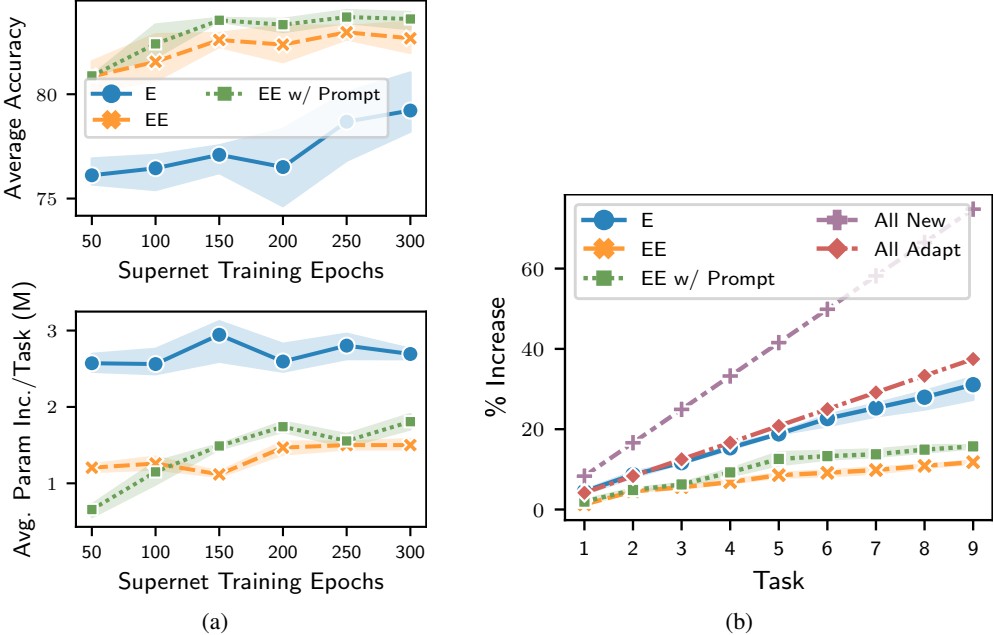

(a)                                                        (b)

Figure 6: (a) Results of the ablation study on the Exploration-Exploitation (EE) guided sampling in the Supernet NAS training using the VDD benchmark (Rebuffi et al., 2017a). The proposed EE sampling strategy is much more efficient than the pure exploration based strategy (i.e., the vanilla SPOS NAS (Guo et al., 2020)). It uses the Supernet training efficiently even at 50 epochs and achieves better performance than pure Exploration, which is desirable for fast adaptation in dynamic environments using lifelong learning. The % increase in parameters shows that EE strategy is effective in reusing experts from the previous tasks and limiting the increase in parameters. The results have been averaged over 3 runs with different seeds. (b) Percent increase in the number of parameters over tasks. All New refers to a new projection layer for every block as new task arrives (similar for All Adapt). This shows a linear increase in the number of parameters. The proposed exploration-exploitation method stays well below the "All Adapt" curve as opposed to pure exploration which almost approaches "All Adapt".

## C.2 EVALUATING FEASIBILITY OF OTHER ViT COMPONENTS AS ARTIHIPPO

Table 6 shows the accuracy with other components if the ViT used for learning the Mixture of Experts using the proposed NAS method. The Query component from the MHSA block and the Value do not perform as well as the Projection layer. The FFN performs only slightly better than the Projection layer, but as a much larger parameter cost This reinforces our identification of the ArtiHippo in Section 3.1 in the main text as a lightweight plastic component.

## C.3 THE EXPLORATION-EXPLOITATION SAMPLING METHOD

Figure 6, left shows that the proposed exploration-exploitation strategy can consistently obtain higher accuracy than pure exploration even when the supernet is trained a small number of epochs. Even when the supernet is trained for a longer duration, the proposed exploration-exploitation strategy still outperforms pure exploration (Figure 6 top). Moreover, the exploration-exploitation strategy adds a lot less additional parameters than pure exploration (Figure 6 bottom). We thus verify that the proposed exploration-exploitation strategy is effective and efficient in utilizing the parameters learned by the previous tasks, thus making the "selective addition" of parameters mentioned in Section 1 possible. This also shows that proposed task-similarity metric is meaningful.

## C.4 PARAMETER GROWTH OVER TIME

Since dynamic model based methods add parameters as new tasks arrive, its necessary to study the rate at which the number of parameters grow. Since the proposed ArtiHippo adds new experts (parameters) dynamically, we cannot analytically determine the rate of growth. However, our experiments show that the proposed exploration-exploitation strategy achieves sub-linear growth in the

| Method-Backbone | ImNet | C100 | SVHN | UCF | OGlt | GTSR | DPed | Flwr | Airc. | DTD | Avg. Accuracy |
|---|---|---|---|---|---|---|---|---|---|---|---|
| S-Prompts[†] ($L$=12) (Wang et al., 2022a) | 82.65 | 89.32 | 88.91 | 64.52 | 72.17 | 99.29 | **99.89** | **96.93** | 45.55 | 60.76 | $80.00 \pm 0.07$ |
| L2P[†] ($L$=12) (Wang et al., 2022d) | 82.65 | 89.32 | 89.89 | 65.63 | 72.34 | 99.55 | **99.94** | **96.63** | 45.24 | 59.57 | $80.08 \pm 0.10$ |
| *L2G (Li et al., 2019) (DARTS) | 82.65 | 88.47 | 85.20 | **79.22** | 80.19 | 99.28 | 98.06 | 76.14 | 39.29 | 46.01 | $77.45 \pm 2.41$ |
| *L2G (Li et al., 2019) ($\beta$-DARTS) | 82.65 | 88.95 | 94.73 | 75.31 | 79.76 | 99.84 | 99.76 | 78.86 | 34.50 | 47.09 | $78.14 \pm 0.54$ |
| EWC (Kirkpatrick et al., 2017) | 58.19 | 87.69 | 69.64 | 57.27 | 45.89 | 95.01 | 98.47 | 90.20 | 36.57 | **61.97** | 70.09 |
| L2 Regularization (Smith et al., 2023b) | 55.28 | 87.10 | 55.23 | 58.86 | 40.48 | 95.07 | 99.17 | 90.20 | 37.53 | **62.55** | 68.15 |
| Experience Replay (Rebuffi et al., 2017b) | 55.88 | 78.70 | 87.40 | 58.20 | 76.03 | 97.92 | 48.55 | 84.41 | 40.98 | 54.68 | 68.27 |
| Our ArtiHippo (Uniform, 150 epochs) | 82.65 | 76.20 | 95.60 | 75.14 | 80.72 | **99.92** | 99.86 | 76.41 | 42.74 | 41.74 | $77.10 \pm 0.75$ |
| Our ArtiHippo (Hierarchical, 50 epochs) | 82.65 | **90.97** | 96.05 | 75.20 | 82.36 | **99.91** | 99.58 | 87.16 | 42.10 | 52.54 | $\mathbf{80.85 \pm 0.72}$ |
| Our ArtiHippo (Hierarchical, 150 epochs) | 82.65 | **90.86** | 96.06 | 75.63 | **84.06** | **99.92** | 99.83 | 89.28 | **51.94** | 55.78 | $\mathbf{82.60 \pm 0.55}$ |
| Our ArtiHippo (Hierarchical+$L$=1, 150 epochs) | 82.65 | **90.50** | **96.19** | **79.70** | **85.71** | **99.91** | 99.83 | 92.42 | **52.23** | 58.99 | $\mathbf{83.55 \pm 0.09}$ |

Table 7: Results on the VDD benchmark (Rebuffi et al., 2017a). Our ArtiHippo shows clear improvements over the previous approaches. All the results from our experiments are averaged over 3 different seeds. The 2 highest accuracies per task have been highlighted. All the methods use the same ViT-B/8 backbone containing 86.04M parameters and having 14.21G FLOPs unless otherwise stated. [†] our modifications for the task-incremental setting. * our reproduction with the vanilla L2G method (Li et al., 2019) for the ViT backbone.

number of parameters, which again shows that our method can effectively leverage the parameters learned in the previous tasks.

## D    COMPARISON WITH ADDITIONAL METHODS

For completeness, we also compare with a baseline of L2 Parameter Regularization (Smith et al., 2023b), and Elastic Weight Consolidation (Kirkpatrick et al., 2017) applied to the linear projection layer of the Multi Head Self-Attention block. We also compare with Experience Replay with a buffer update strategy used in Rebuffi et al. (2017b). This comparison is not completely fair, since these methods are class-incremental in nature. However, this comparison serves as a baseline to observe meaningful trade-off. Table 7 shows that EWC, L2 Parameter Regularization and Experience Replay cannot completely overvome catastraophic forgetting, and hence lose accuracy over time.

## E    DISCUSSION ON OTHER TRANSFORMER-BASED METHODS

Due to space limitations, we discuss similarity and differences with other transformer-based methods here. Our method follows the similar principle of leveraging the stable and expressive heavy-weight components of a backbone Vision Transformer (ViT) as prompt-based methods Wang et al. (2022d;c;a); Smith et al. (2023a). However, in contrast to prompt-based methods, we choose a lightweight component of the ViT - the final linear projection layer in the Multi-Head Self-Attention (MHSA) block - and (selectively) add additional parameters for new tasks. In our experiments, we show that this strategy leads to higher performance in terms of average accuracy, and is essential when the distribution of the new task is significantly different from the task on which the backbone is trained. Moreover, we show that prompt-based methods are complementary to our proposed method, and combining them leads to an even higher performance.

Other methods which use a ViT in a lifelong learning setting (without prompts) can largely be divided into two categories: methods that use a pretrained backbone, and methods that start from a randomly initialized ViT. We will refer to thse methods as parameter-based methods for simplicity. Xue et al. (2022) use a backbone ViT pretrained on ImageNet and learn binary masks to enable/disable parameters in the Feedforward Network (FFN), and attention between image tokens for downstream tasks. Ermis et al. (2022) train adapters Houlsby et al. (2019) per task, and distill into the most similar adapter after the number of adapters exceed a predefined budget. Iscen et al. (2022) uses a memory bank and a fixed (pretrained) feature extractor, and models the relationship between query features and features from the memory bank using a transformer. Gao et al. (2023) combine LoRA Hu et al. (2021), Adapters Houlsby et al. (2019), and prefix tuning Li & Liang (2021) and use scaling factors to calibrate the learning speeds of prefixes and adapters. They further use a "slow-learner" in the form of an exponentially moving average of the learned parameters of the current task and the model learned on the previous tasks, and uses an ensemble of both models at test time to avoid using task IDs.

Pelosin et al. (2022) use a regularization loss on the attention maps in the MHSA block to prevent catastrophic forgetting. Yu et al. (2021) use a convolutional stem, bias correction and higher

learning rate for the classification head to address underfitting and biases in the classifier Wu et al. (2019). Wang et al. (2022b) uses external task keys and attention biases along with a knowledge distillation objective on the samples from a replay buffer. Douillard et al. (2022) uses a separate task-attention module on top of a transformer-based feature extractor, and use a memory buffer to counter catastrophic forgetting. Mohamed et al. (2023) uses a hybrid ViT architecture along with Class Activation Maps (CAM) of the attention mechanism for distillation of the previous task data.

Our approach uses the first task as Imagenet Russakovsky et al. (2015), similar to prompt based methods Wang et al. (2022d;c;a); Smith et al. (2023a), and parameter based methods like Xue et al. (2022); Ermis et al. (2022); Iscen et al. (2022) that use a pretrained backbone model. Our approach differs from other parameter-based approaches in that we identify a "sweet-spot" in the ViT architecture and selectively add parameters only to the final projection layer in the MHSA block. While Ermis et al. (2022) is similar, we follow the opposite philosophy: we do not keep a fixed budget on the number of additional parameters that we can add, but control the rate of addition through our exploration-exploitation driven strategy. Figure 6 shows that our exploration-exploitation strategy indeed prevents an unrestricted addition of parameters. See Section C.3 and Section C.4 for details. The main difference between Gao et al. (2023) and our proposed ArtiHippo is that our method (based on the growing operations like Learn to Grow Li et al. (2019)) can explicitly reuse some experts, whereas Gao et al. (2023) tune all the layers and try to preserve stability by adapting the learning speed of the newly added components.

## F    Modifying dynamic model based methods for Vision Transformers

Supermasks in Superposition (SupSup, (Wortsman et al., 2020)), Efficient Feature Transformation (EFT, (Verma et al., 2021)), and Lightweight Learner (LL, (Ge et al., 2023b)) have originally been developed for Convolutional Neural Networks. Here, we describe our modifications to theoriginal methods to make them compatible with Vision Transformers for a fair comparison with our Arti-Hippo. Following ArtiHippo, which learns a base Vision Transformer model with ImageNet data from the VDD benchmark (Rebuffi et al., 2017a), we initialize the network for SupSup, EFT and LL with the same backbone. For SupSup, we learn masks for the weights of the final linear projection layer of the Multi-Head Self-Attention block using the straight through estimator (Bengio et al., 2013). We apply EFT on all the linear layers of the ViT by scaling all the activations by a learnable scaling vector using the Hadamard product following the original proposed formulation for fully-connected layers. Finally, for LL, which learns a task-specific bias vector which is added to all the feature maps of convolutional layers, we learn a similar bias vector and add it to the output of all the linear layers of the ViT.

## G    Modifying S-Promts and L2P for task-incremental setting

Both, Learn to Prompt and S-Prompts, can be modified for task-incremental setting without altering the core algorithm for learning the prompts. For S-Prompts, this is done by training a separate prompt of length $L$, i.e., (i.e. a $L$ cls tokens) per task and retrieving the correct prompt using the task ID. For L2P, we follow the official implementation[1] used for evaluating on the 5-datasets benchmark. L2P first trains a set of $N$ prompts of length $L_p$ (i.e. $NL_p$ tokens) per task. It then learns a set of $N$ keys such that the distance between the keys and the image encoding (using a fixed feature extractor) is maximized. We retrieve the retrieve the correct prompts using the Task ID instead of using a key-value matching and make L2P compatible with a task-incremental setting. We initialize the the values for the prompts for task $t$ from the trained values of task $t-1$ following the original implementation.

### G.1    Ablation study with varying number of prompts

Table 8 and Table 9 show the results of varying the number of prompt tokens for S-Prompts[†] on the VDD and 5-dataset benchmarks respectively. Varying the number of prompts beyond 10 does not affect the performance significantly, as observed in the original paper (Wang et al., 2022a). Hence,

---

[1]L2P official implementation: https://github.com/google-research/l2p

| Method | ImNet | C100 | SVHN | UCF | OGlt | GTSR | DPed | Flwr | Airc. | DTD | Avg. Accuracy |
|---|---|---|---|---|---|---|---|---|---|---|---|
| S-Prompts† (p=1/task) | 82.65 | 87.06 | 76.42 | 54.82 | 62.10 | 96.74 | 99.59 | 95.52 | 37.62 | 57.78 | $75.03 \pm 0.19$ |
| S-Prompts† (p=5/task) | 82.65 | 88.91 | 85.23 | 62.23 | 70.64 | 99.08 | 99.87 | 97.35 | 45.32 | 60.74 | $79.20 \pm 0.53$ |
| S-Prompts† (p=10/task) | 82.65 | 89.62 | 88.69 | 65.20 | 72.18 | 99.37 | 99.91 | 97.06 | 45.17 | 60.94 | $80.08 \pm 0.30$ |
| S-Prompts† ($L$=12) (Wang et al., 2022a) | 82.65 | 89.32 | 88.91 | 64.52 | 72.17 | 99.29 | 99.89 | 96.93 | 45.55 | 60.76 | $80.00 \pm 0.07$ |
| S-Prompts† (p=15/task) | 82.65 | 89.63 | 89.36 | 65.88 | 72.54 | 99.37 | 99.94 | 97.03 | 45.07 | 61.17 | $80.26 \pm 0.09$ |
| L2P† ($L$=12) (Wang et al., 2022d) | 82.65 | 89.32 | 89.89 | 65.63 | 72.34 | 99.55 | 99.94 | 96.63 | 45.24 | 59.57 | $80.08 \pm 0.10$ |

Table 8: Results on the VDD benchmark Rebuffi et al. (2017a) with various number of prompts. Increasing the number of prompts above 10 does not lead to a significant gain in accuracy.

| Method | Num. Prompts | Avg. Acc. |
|---|---|---|
| S-Prompts† | 1 | $88.93 \pm 0.34$ |
| S-Prompts† | 5 | $91.14 \pm 0.78$ |
| S-Prompts† | 10 | $92.28 \pm 0.16$ |
| S-Prompts† | 12 | $92.42 \pm 0.11$ |
| S-Prompts† | 15 | $92.39 \pm 0.05$ |
| L2P† | 12 | $92.73 \pm 0.10$ |

Table 9: Results on the 5-Dataset benchmark Ebrahimi et al. (2020). The results have been averaged over 5 different task orders following L2P.

for L2P†, we calculate the number of prompt tokens by scaling the number of prompts used in the original implementation of L2P (Wang et al., 2022d) on the 5-datasets benchmark to suit an image size of $72 \times 72$. For the 5 datasets benchmark, L2P uses 4 prompts of length 10 (40 tokens) for an image size of $224 \times 224$. We scale the prompt length to `floor`$(10/3) = 3$, which gives 4 prompts of length 3, i.e., 12 tokens. We use a scaling factor of 3 since 224/72=3.

## H   DATASET DETAILS

### H.1   THE VDD BENCHMARK

It consists of 10 tasks: ImageNet12 (Russakovsky et al., 2015), CIFAR100 (Krizhevsky et al., 2009), SVHN (Netzer et al., 2011), UCF101 Dynamic Images (UCF) (Soomro et al., 2012; Bilen et al., 2016), Omniglot (Lake et al., 2015), German Traffic Signs (GTSR) (Stallkamp et al., 2012), Daimler Pedestrian Classification (DPed) (Munder & Gavrila, 2006), VGG Flowers (Nilsback & Zisserman, 2008), FGVC-Aircraft (Maji et al., 2013), and Describable Textures (DTD) (Cimpoi et al., 2014). All the images in the VDD benchmark have been scaled such that the shorter side is 72 pixels. Table 10 shows the number of samples in each task. Figure 7 shows examples of images from each task of the VDD benchmark.

In our experiments, we use 10% of `the official training data` from each of the tasks for validation (e.g., used in the target network selection in Section 3.2.3 in main text), and report the accuracy on `the official validation set` for fair comparison with the learn-to-grow method (Li et al., 2019) in Table 2. In Table 10, the `train, validation` and `test` splits are thus referred to 90% of the official training data, 10% of the official training data, and the entire official validation data respectively. When finetuning the learned architecture, we use the entire `the official training data` to train and report results on the `the official validation set`.

### H.2   THE 5-DATASETS BENCHMARK

It consists of 5 tasks: CIFAR10 (Krizhevsky et al., 2009), MNIST (Lecun et al., 1998), Fashion-MNIST (Xiao et al., 2017), not-MNIST (Bulatov, 2011), and SVHN (Netzer et al., 2011). Table 11 shows the data statistics. Figure 8 shows examples of images from each task. To be consistent with the settings used on the VDD benchmark, we use 15% of `the training data` for validation and report the results on `the official test data`, except for not-MNIST for which an official test split is not available. So, for the not-MNIST dataset, we use the small version of that dataset, with which we construct the test set by randomly sampling 20% of the samples. From the remaining 80%, we use 15% for validation, and the rest as the training set.

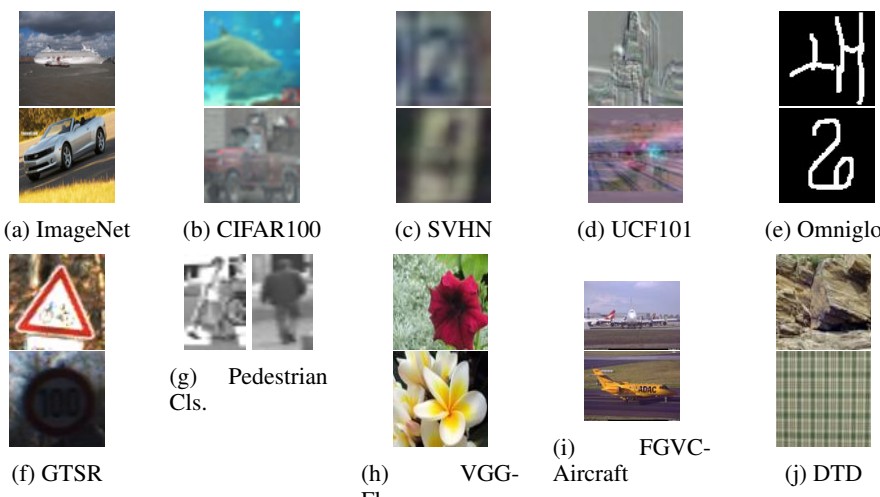

Figure 7: Example images from the VDD benchmark (Rebuffi et al., 2017a). Each task has a significantly different domain than others, making VDD a challenging benchmark for lifelong learning.

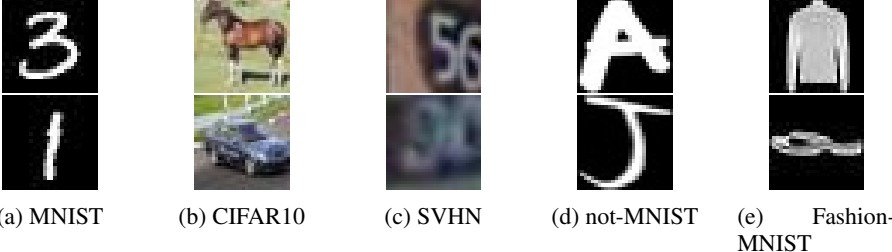

Figure 8: Example images from the 5-Datasets benchmark (Ebrahimi et al., 2020).

| Task | Train | Validation | Test |
|------|-------|------------|------|
| ImageNet12 | 1108951 | 123216 | 49000 |
| CIFAR100 | 36000 | 4000 | 10000 |
| SVHN | 42496 | 4721 | 26040 |
| UCF | 6827 | 758 | 1952 |
| Omniglot | 16068 | 1785 | 6492 |
| GTSR | 28231 | 3136 | 7842 |
| DPed | 21168 | 2352 | 5880 |
| VGG-Flowers | 918 | 102 | 1020 |
| Aircraft | 3001 | 333 | 3333 |
| DTD | 1692 | 188 | 1880 |

Table 10: The number of samples in training, validation and testing sets per task used in our experiments on the VDD benchmark (Rebuffi et al., 2017a).

| Task | Train | Validation | Test |
|------|-------|------------|------|
| MNIST | 51000 | 9000 | 10000 |
| not-MNIST | 12733 | 2247 | 3744 |
| SVHN | 62269 | 10988 | 26032 |
| CIFAR10 | 42500 | 7500 | 10000 |
| Fashion MNIST | 51000 | 9000 | 10000 |

Table 11: Number of samples in training, validation, and test sets per task in the 5-Datasets benchmark (Ebrahimi et al., 2020). The test samples have been reported from the official test data provided by each individual dataset, except for not-MNIST. See text for details.

## I  THE VISION TRANSFORMER: VIT-B/8

We use the base Vision Transformer (ViT) model, with a patch size of $8 \times 8$ (ViT-B/8) model from (Dosovitskiy et al., 2021). The base ViT model contains 12 Transformer blocks with residual connections for each block. A Transformer block is defined by stacking a Multi-Head Self-Attention (MHSA) block and a Multi-Layer Perceptron (MLP) block with resudual connections for each block. ViT-B/8 uses 12 attention heads in each of the MHSA blocks, and a feature dimension of 768. The MLP block expands the dimension size to 3072 in the first layer and projects it back to 768 in the second layer. For all the experiments, we use an image size of $72 \times 72$ following the VDD setting. We base the implementation of the ViT on the `timm` package (Wightman, 2019).

### I.1 IMAGENET TRAINING ON THE VDD BENCHMARK

To train the ViT-B/8 model, we use the ImageNet data provided by the VDD benchmark (the `train` split in Table 10). To save the training time, we initialize the weights from the ViT-B/8 trained on the full resolution ImageNet dataset (224×224) and available in the `timm` package, and finetune it for 30 epochs on the downsized version of ImageNet (72×72) in the VDD benchmark. We use a batch size of 2048 split across 4 Nvidia Quadro RTX 8000 GPUs. We follow the standard training/finetuning recipes for ViT models. The file `artihippo/artifacts/imagenet_pretraining/args.yaml` provides all the training hyperparameters used for training the the ViT-B/8 model on ImageNet. During testing, we take a single center crop of 72×72 from an image scaled with the shortest side to scaled to 72 pixels.

## J BACKGROUND: LEARN TO GROW AND SPOS

### J.1 BACKGROUND ON LEARN TO GROW

The learn-to-grow method (Li et al., 2019) uses Differentiable Architecture Search (DARTS) (Liu et al., 2019a), a supernet based NAS algorithm to learn a strategy for reusing, adapting, or renewing the parameters learned for the previous tasks (skipping is not applied). Consider an $L$-layer backbone network with $\mathcal{S}^l$ choice of parameters for a layer $l$ learned for the previous tasks. The parameters can be the weights and biases of the layer in the case of fully-connected layers, or the filters and the biases in the case of a Convolutional layers. For a new task, the learn-to-grow method constructs the search space for NAS (referred to as operations) by applying the operations `reuse`, `adapt`, and `new` to $\mathcal{S}^l$ for all layers $l \in [1, L]$. The total number of choices for layer $l$ is $C_l = 2|\mathcal{S}^l| + 1$ $\{|\mathcal{S}^l|$ reuse, $|\mathcal{S}^l|$ adapt, and 1 new operation$\}$. Following DARTS, the output of a layer when training the NAS supernet is given by

$$x_{out}^l = \sum_{c=1}^{C_l} \frac{exp(\alpha_c^l)}{\sum_{c'=1}^{C_l} exp(\alpha_{c'}^l)} g_c^l(x_{in}^l), \tag{6}$$

where $g_c^l(\cdot)$ is defined by,

$$g_c^l(x_{l-1}) = \begin{cases} S_c^l(x_{in}^l) & \text{if } c \leq |\mathcal{S}^l|, \\ S_c^l(x_{in}^l) + \gamma_{c-|\mathcal{S}^l|}^l(x_{in}^l) & \text{if } |\mathcal{S}^l| < c \leq 2|\mathcal{S}^l|, \\ new^l(x_{in}^l) & \text{if } c = 2|\mathcal{S}^l| + 1, \end{cases} \tag{7}$$

where $\gamma$ denotes the additional operation used in parallel ($1 \times 1$ convolution in the case of a Convolutional layer) which implements the `adapt` operation. Using DARTS, the operations are trained jointly with architecture coefficients $\alpha_c^l$. Once the supernet is trained, the optimal operation is selected such that $(c^*)^l \leftarrow \text{argmax}_c \ \alpha_c^l$. In experiments, the learn-to-grow method uses a 26-layer ResNet (He et al., 2016) on the VDD benchmark.

**Remarks on the proposed learning-to-grow with ViTs.** The learn-to-grow method (Li et al., 2019) can maintain dynamic feature backbone networks for different tasks based on NAS, which leads to the desired selectivity and plasticity of networks in lifelong learning. It has been mainly studied with Convolutional Neural Networks (CNNs), and often apply DARTS for all layers with respect to the three operations, which is time consuming and may be less effective when a new task has little data.

More importantly, the vanilla learn-to-grow method with DARTS and ResNets does not have the motivation of integrating learning and (long-term) memory in lifelong learning, which is the focus of our proposed ArtiHippo method. In learning a new task, unlike the vanilla learn-to-grow method in which all the previous tasks are treated equally when selecting the operation operations, our proposed ArtiHippo leverages the task similarities which in turn exploits the class-token specialized in ViTs.

### J.2 BACKGROUND ON THE SINGLE-PATH ONE-SHOT NEURAL ARCHITECTURE SEARCH METHOD

DARTS (Liu et al., 2019a) trains the entire supernet jointly in learning a new task, and thus might not be practically scalable and sustainable after the supernet "grew too fat" at each layer. The strategy

used in Single-Path One-Shot (SPOS) (Guo et al., 2020) NAS offers an alternative strategy based on a *stochastic* supernet. It uses a bi-level optimization formulation consisting of the supernet training and the target network selection,

$$W_{\mathcal{A}} = \text{argmin}_W \mathcal{L}_{train}(\mathcal{N}(\mathcal{A}, W)), \tag{8}$$

$$a^* = \text{argmax}_{a \in \mathcal{A}} \, Acc_{val}(\mathcal{N}(a, W_{\mathcal{A}}(a))), \tag{9}$$

where Eqn. 8 is solved by defining a prior distribution $\Gamma(\mathcal{A})$ over the choice of operations in the stochastic supernet and optimizing,

$$W_{\mathcal{A}} = \text{argmin}_W \mathbf{E}_{a \sim \Gamma(\mathcal{A})} \mathcal{L}_{train}(\mathcal{N}(\mathcal{A}, W)). \tag{10}$$

This amounts to sampling one operation per layer (i.e., one-shot) of the neural network, and to forming a single path in the stochastic supernet. Eqn. 9 is optimized using an evolutionary search based on the validation performance for different candidates of the target network in a population, which is efficient since only inference is executed. The SPOS method empirically finds that a uniform prior works well in practice, especially when sufficient exploration is afforded. Note that this prior is also applied in generating the initial population for the evolutionary search.

The evolutionary search method used in the SPOS method is adopted from (Real et al., 2019). It first initializes a population with a predefined number of candidate architectures sampled from the supernet. It then "evolves" the population via the crossover and the mutation operations. At each "evolving" iteration, the population is evaluated and sorted based on the validation performance. With the top-$k$ candidates after evaluation and sorting (the number $k$ is predefined), for crossover, two randomly sampled candidate networks in the top-$k$ are crossed to produce a new target network. For mutation, a randomly selected candidate in the top-$k$ mutates its every choice block with probability (e.g., $0.1$) to produce a new candidate. Crossover and mutation are repeated to generate sufficient new candidate target networks to form the population for the next "evolving" iteration.

**Remarks on the proposed hierarchical task-similarity-oriented sampling with exploitation-exploitation trade-off.** We modify the core sampling component in the SPOS algorithm for resilient lifelong learning with a long-term memory. During exploitation, we generate the prior $\Gamma(\mathcal{A})$ using our proposed hierarchical sampling scheme, and use the uniform prior during exploration. The same sampling scheme is applied when generating the initial population for the evolutionary search as well.

## K    SETTINGS AND HYPERPARAMETERS IN THE PROPOSED LIFELONG LEARNING

Starting with the ImageNet pretrained ViT-B/8, the proposed lifelong learning methods consists of three components in learning new tasks continually and sequentially: supernet training, evolutionary search for target network selection, and target network finetuning. The supernet training and target network finetuning use the `train` split, while the evolutionary search uses the `validation` split, both shown in Table 10 and Table 11. We use the vanilla data augmentation in both supernet training and target network finetuning. We use a weight of 1 for the beta loss in all the experiments with $\beta$-DARTS.

| Task | Scale and Crop | Hor. Flip | Ver. Flip |
|------|----------------|-----------|-----------|
| CIFAR100 | Yes | p=0.5 | No |
| Aircraft | Yes | p=0.5 | No |
| DPed | Yes | p=0.5 | No |
| DTD | Yes | p=0.5 | p=0.5 |
| GTSR | Yes | p=0.5 | No |
| OGlt | Yes | No | No |
| SVHN | Yes | No | No |
| UCF101 | Yes | p=0.5 | No |
| Flwr. | Yes | p=0.5 | No |

Table 12: Data augmentations for the 9 tasks in the VDD benchmark.

| Task | Scale and Crop | Hor. Flip |
|------|----------------|-----------|
| MNIST | Yes | No |
| not-MNIST | Yes | No |
| SVHN | Yes | No |
| CIFAR100 | Yes | p=0.5 |
| Fashion MNIST | Yes | No |

Table 13: Data augmentations used for each task in the 5-Datasets benchmark.

**Data Augmentations.** A full list of data augmentations used for the VDD benchmark is provided in Table 12, and the data augmentations used for the tasks in the 5-datasets benchmark is provided in Table 13. The augmentations are chosen so as not to affect the nature of the data. Scale and Crop transformation scales the image randomly between 90% to 100% of the original resolution and takes a random crop with an aspect ratio sampled from a uniform distribution over the original

aspect ratio $\pm 0.05$. In evaluating the supernet and the finetuned model on the validation set and test set respectively, images are simply resized to $72 \times 72$ with bicubic interpolation.

## K.1 SUPERNET TRAINING

*VDD Benchmark*: For each task, we train the supernet for 150 epochs, unless otherwise stated. We use a label smoothing of 0.1. No other form of regularization is used since the `skip` operation provides implicit regularization, which plays the role of Drop Path during training. We use a learning rate of 0.001 and the Adam optimier (Kingma & Ba, 2014) with a Cosine Decay Rule. For experiments with separate class tokens per task, we use a learning rate of 0.0005 for training the supernet, and 0.001 for training the task token. For each epoch, a minimum of 15 batches are drawn, with a batch size of 512. If the number of samples present in the task allows, we draw the maximum possible number of batches that covers the entire training data. For the Exploration-Exploitation sampling scheme, we use an exploration probability $\epsilon = 0.3$.

*5-datasets Benchmark*: We use the same hyperparameters as those used in the VDD Benchmark, but train the supernet for 50 epochs.

*L2G with DARTS and $\beta$-DARTS*: We train the supernet for Learn to Grow (Li et al., 2019) for 50 epochs for the VDD benchmark and 25 epochs for the 5-datasets benchmark.

## K.2 EVOLUTIONARY SEARCH

The evolutionary search is run for 20 epochs. We use a population size of 50. 25 candidates are generated by mutation, and 25 candidates are generated using crossover. The top 50 candidates are retained. The crossover is performed among the top 10 candidates, and the top 10 candidates are mutated with a probability of 0.1. For the Exploration-Exploitation sampling scheme, we use an exploration probability $\epsilon = 0.5$ when generating the initial population.

## K.3 FINETUNING

The target network for a task selected by the evolutionary search is finetuned for 30 epochs with a learning rate of 0.001, Adam optimizer, and a Cosine Learning Rate scheduler. Drop Path of 0.25 and label smoothing of 0.1 is used for regularization. We use a batch size of 512, and a minimum of 30 batches are drawn. When using a separate task token for each task, the task token is first finetuned with a learning rate of 0.001.

## K.4 NORMALIZED COSINE SIMILARITY

To verify the use of the Normalized Cosine Similarity as our similarity measure, we refer to Figure 9. Figure 9a shows the Cosine Similarity between the mean class-tokens learned for tasks ImageNet, CIFAR100, SVHN, UCF101, and Omniglot (in order), and the mean class-tokens calculated for each expert using the data from the current task GTSR. Empirically, we observe that the Cosine Similarity between the mean class-tokens calculated using the data of the task associated with an expert and the mean class-token calculated with the current task in training is high. However, the difference between the similarity values for each expert are more important than the absolute values of the similarity. This difference can be increased by scaling the similarity such that it increases the magnitude difference between the similarities of different tasks, but maintains the relative similarity. This can be achieved by scaling the Cosine Similarities between -1 and 1 using the minimum and the maximum values from all the experts and all the blocks (Figure 9b). Using the Normalized Cosine Similarity leads to better and more intuitive probability distributions for sampling candidate experts and the retention probabilities for the sampled experts. For example, comparing the probability values for sampling an expert at Block 6 calculated using Cosine Similarity (Figure 9c) vs. Normalized Cosine Similarity (Figure 9d), the probability of sampling the ImageNet expert increases, and those of sampling UCF101 and Omniglot decrease. Similarly, for Blocks 5 and 6, the retention probability calculated using the Normalized Cosine Similarity (Figure 9f) reduces by a large factor than that calculated using the Cosine Similarity (Figure 9e). This will encourage sampling the `adapt` operation when these experts are sampled, thus adding plasticity to the network. The retention probability of the ImageNet experts at these blocks also reduces slightly, which will avoid imposing a strict prior.

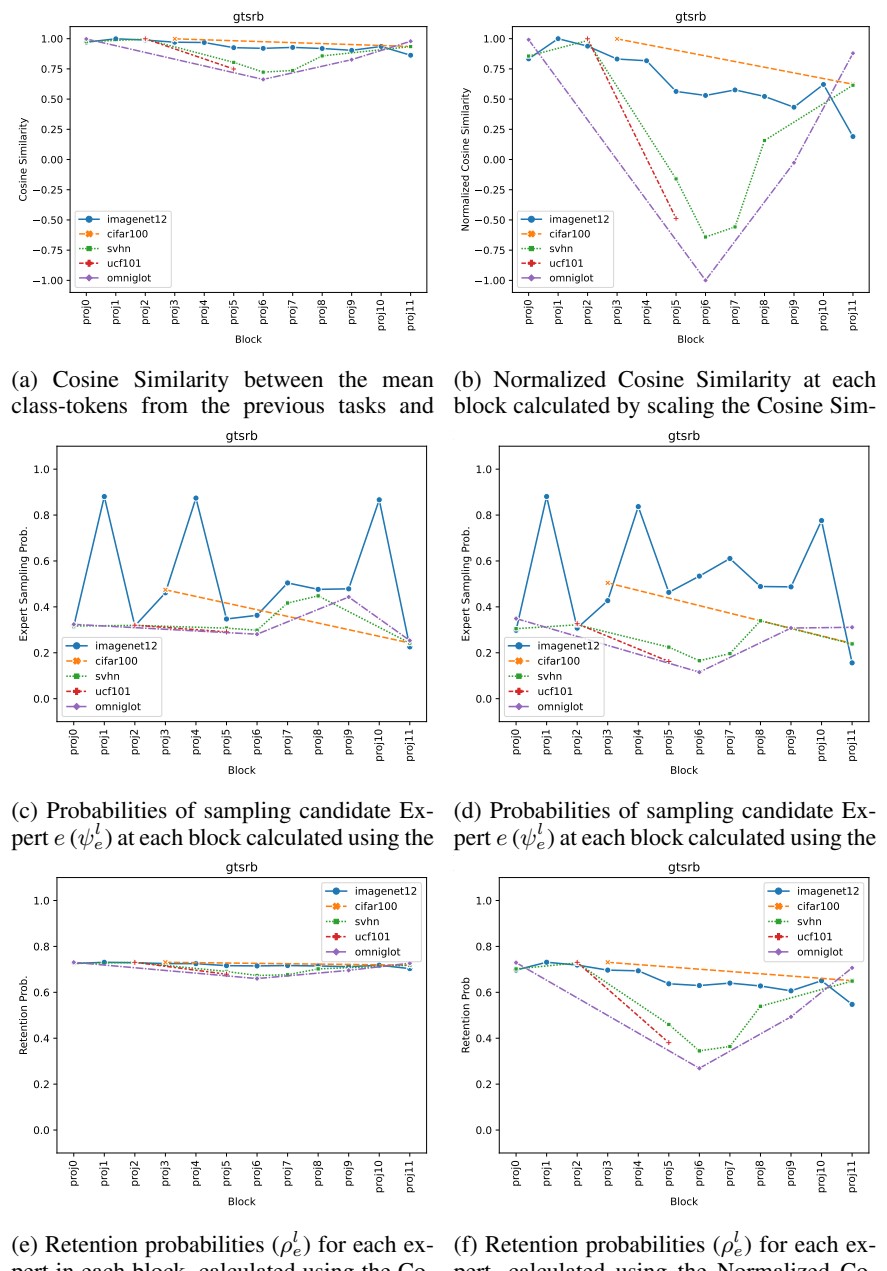

(a) Cosine Similarity between the mean class-tokens from the previous tasks and

(b) Normalized Cosine Similarity at each block calculated by scaling the Cosine Sim-

(c) Probabilities of sampling candidate Expert $e$ $(\psi_e^l)$ at each block calculated using the

(d) Probabilities of sampling candidate Expert $e$ $(\psi_e^l)$ at each block calculated using the

(e) Retention probabilities $(\rho_e^l)$ for each expert in each block, calculated using the Cosine Similarity.

(f) Retention probabilities $(\rho_e^l)$ for each expert, calculated using the Normalized Cosine Similarity.

Figure 9: Comparison of probability values for sampling the Experts (Middle row) and the retention probabilities (Bottom row) using the Cosine Similarity and the Normalized Cosine Similarity. Using the Normalized Cosine Similarity gives better probability values for expert sampling probabilities $\psi_e$, as seen in Blocks 5 and 6, where the expert sampling probability for ImageNet increases, thus reducing the probability of sampling `new` and `skip`. This will encourage maximal reuse. The effect on the retention probability $\rho_e$ can be prominently seen on the Omniglot experts. The retention probability in Blocks 6 9f reduces, which will encourage the `adapt` operation to be trained even if the Omniglot expert even if Omniglot experts were sampled.

## L LEARNED ARCHITECTURE FOR DIFFERENT TASK ORDER AND PURE EXPLORATION

Figure 10 shows the architecture learned for a different task sequence. It can be seen that even with a different task sequence, the proposed method can learn to exploit task similarities. For example, an adapt operation is layer is learned at Block 4 for Omniglot, which is reused by GTSR. At Block 7, a New operation is learned for Omniglot, which is adapted for SVHN. Even though CIFAR100 is learnt as the last task, the search process can still learn to reuse many ImageNet experts. Figure 11 shows the architecture learned using a pure exploration strategy. It can be seen that pure exploration does not reuse components from similar tasks. For example, a large number of `Adapt` and `New` are added when learning CIFAR100. In contrast, the exploitation-exploitation strategy can learn to reuse the components from the ImageNet task (Figure 5 in the main text), and achieves better accuracy as well (Table 2 in the main text).

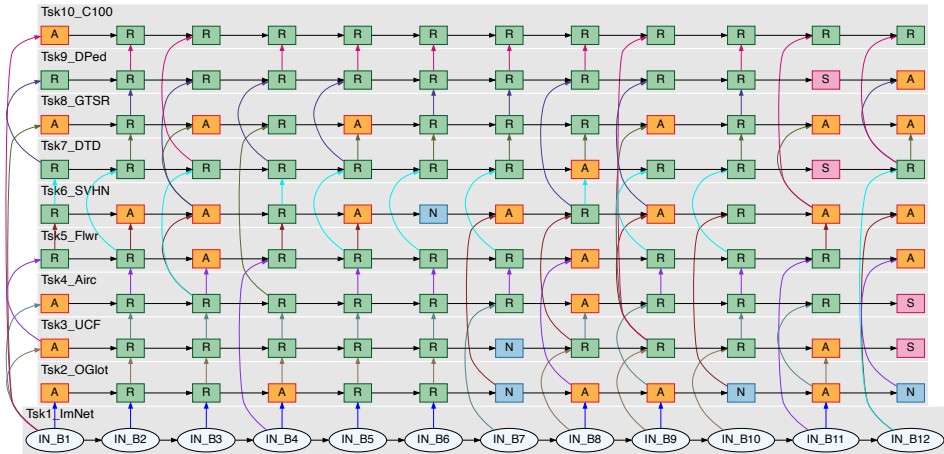

Figure 10: Architecture learned for task sequence ImNet, OGlt, UCF, Airc, Flwr, SVHN, DTD, GTSR, DPed, C100

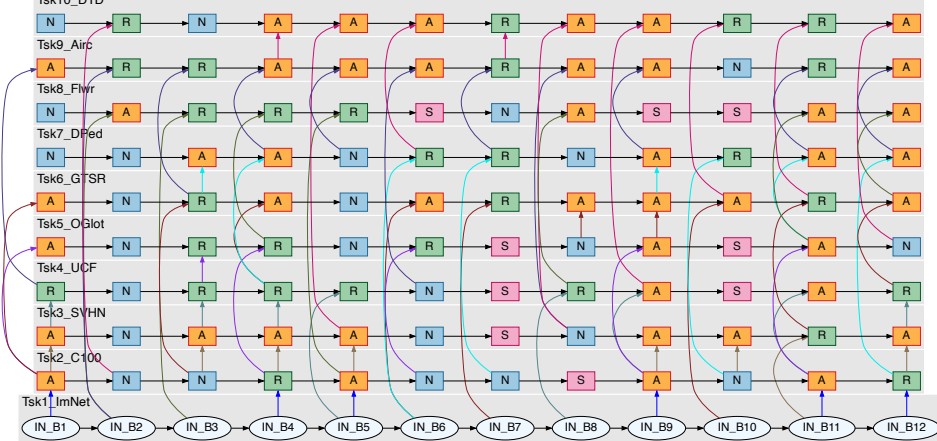

Figure 11: Architecture learned using pure exploration. Pure Exploration based method adds many unnecessary Adapt and New operations even though the tasks are similar (ImNet $\rightarrow$ C100), proving the effectiveness of the proposed sampling method

