# OpenReview forum: "Transforming Transformers for Resilient Lifelong Learning"
_ICLR.cc/2024/Conference — Submitted to ICLR 2024_

### Official Review · Reviewer_Z1nR · 2023-10-29

**Soundness:** 3 good
**Presentation:** 3 good
**Contribution:** 3 good
**Rating:** 6
**Confidence:** 4

**Summary:**

The paper introduces a method for achieving resilient lifelong learning in deep neural networks, focusing on Transformers, which have gained prominence in deep learning. The key challenge addressed is catastrophic forgetting, where networks struggle to learn new tasks without losing previously acquired knowledge.

The approach suggests integrating task-aware, adaptable components (referred to as Artificial Hippocampi or ArtiHippo) into the architecture of Vision Transformers (ViTs).

Inspired by the human brain's Hippocampi, known for their role in lifelong learning, the paper explores how these artificial components can be identified, placed, and trained within ViTs to enable adaptability while preserving core functions.

The proposed method belongs to the parameter-tuning category with more fine-grained control.

**Strengths:**

This paper presents an interesting concept "Artificial Hippocampi (ArtiHippo) in Transformers" for task-incremental learning. The proposed method introduces lightweight transformer components to the dynamic networks to learn the new tasks.

The paper is written and organized well.

It obtains better performance than the prior art with sensible ArtiHippo learned continually.

**Weaknesses:**

The approach proposed in this article needs to rely on a priori, i.e., knowing in advance to which task a certain data belongs. This limits the practical application scenarios of the CL approach.
In addition, the method does not scale well as the number of tasks increases. Especially for scenarios where multiple small tasks exist.

Lack of experiments on generic experimental datasets, inclusion of ImageNet and CIFAR100.

Lack of comparison with enough continual learning methods, please discuss or compare with recent proposed baselines.

Missing related Transformer-based continual learning methods, e.g.,

[1] Continual Learning with Transformers for Image Classification. CVPR 2022 CLVision workshop

[2] Continual Learning with Lifelong Vision Transformer. CVPR 2022

[3] D3Former: Debiased Dual Distilled Transformer for Incremental Learning. CVPR 2023

**Questions:**

Why Tokenized Data is used as input data in Figure 1, is there any special meaning?

Please discuss the differences and similarities between this paper with other Transformer-based CL methods.

---

> ### Author Response · Authors · 2023-11-22
> **Rebuttal by Authors**
>
> Thank you for your time and efforts reviewing our paper. We address your concerns as follows.
>
> ### Limitations of the method: task incremental setting
>
> We acknowledge that the task incremental setting is a limitation of our method. We note that the class incremental setting has its own limitations, e.g., it often requires the number of classes between different tasks is the same, which is not applicable in the VDD benchmark we tested. In the meanwhile, the task incremental setting remains a commonly used setting in studying lifelong learning, where the focus is how to achieve maximal forward transfer across tasks that may have different output space by reusing the relevant parameters learned for all the previous tasks and adding minimal new parameters, which is also an equally important problem.
>
> ### Limitations: scaling to larger number of tasks
>
> We agree and have acknowledged both of these limitations in the submission. However, we would like to point out that our proposed exploration-exploitation sampling scheme can perform well with just 50 epochs of supernet training, whereas pure exploration cannot match this accuracy even with 300 epochs. This shows that our method can scale better than pure exploration. However, very few benchmarks containing a larger number of tasks exist that can tackle our goal of testing the behavior of prompt-based methods and our proposed parameter-based method on diverse tasks on our computational budget (with SKILL benchmark [4] being a very recent development). We will address this issue in future work.
>
> ### Lack of experiments on generic experimental datasets, inclusion of ImageNet and CIFAR100.
>
> One of the goals of work is to study the behavior of prompt-based methods and parameter based (methods which add/train the network parameter) methods on diverse tasks. This diversity can come from a change in the input distribution and/or output distribution. Hence, VDD is an appropriate benchmark for this study, since it offers a wide diversity in tasks: Natural images (flowers, aircraft, CIFAR100), street view images (SVHN, GTSR), handwritten digits (omniglot), etc. Since CIFAR100 is already present in the VDD benchmark, it is more generic and challenging than CIFAR100. Furthermore, Learn to Prompt and S-Prompt use a model pretrained on ImageNet, and hence a evaluating on ImageNet will not be an appropriate experiment.
>
> ### Lack of comparison with enough continual learning methods, please discuss or compare with recent proposed baselines.
>
> We have compared with recently proposed baselines that offer a fair comparison with our method, i.e., that start from a model trained in ImageNet: Learn to Prompt (CVPR22), S-Prompts (NeurIPS22) and Lightweight Learner (TMLR23). Please let us know if there are any specific references that use similar settings.
>
> ### Why Tokenized Data is used as input data in Figure 1, is there any special meaning?
>
> Figure 1 shows the typical lifelong learning methods that use the Transformer architecture. Since Transformers handle the input data by the means of tokenization (e.g. text tokens of image tokens), we represent the data in the form of tokens, and make a distinction between data tokens and prompt tokens in Figure 1a.
>
> ### Missing related work, similarities and differences with other Transformer based continual learning methods
>
> Thank you for pointing put these methods to us. We will cite these and discuss the differences in the Appendix of the revised submission due to space limitations. Please note that we have cited [1] (last reference on page 11 in the initial submission)
>
> > [1] Continual Learning with Transformers for Image Classification. CVPR 2022 CLVision workshop
>
> > [2] Continual Learning with Lifelong Vision Transformer. CVPR 2022
>
> > [3] D3Former: Debiased Dual Distilled Transformer for Incremental Learning. CVPR 2023
>
> > [4] Lightweight Learner for Shared Knowledge Lifelong Learning. TMLR 2023

---

### Official Review · Reviewer_6RhY · 2023-10-29

**Soundness:** 2 fair
**Presentation:** 3 good
**Contribution:** 2 fair
**Rating:** 6
**Confidence:** 4

**Summary:**

This paper introduces a method of training vision transformers (ViTs) for lifelong learning under the task-incremental setting. It identifies the final projection layer of multi-head self-attention of a ViT as the Artificial Hippocampi (ArtiHippo) of ViTs, and learns to dynamically grow the ArtiHippo by four operations "Skip", "Resue", "Adapt" and "New". The maintenance of ArtHippo is realized by hierarchical exploration-exploitation sampling where the exploitation utilizes task similarities measured by the normalized cosine similarity between the mean class tokens of a new task and those of old tasks. Experiments are conducted on VDD and 5-Datasets benchmarks, showing better performance than previous art.

**Strengths:**

1. The paper is clearly written, well-presented with rich visualizations and easy to follow.
2. The method is effective when utilizing ViTs for lifelong learning.
3. The analysis of results is thorough and insightful.

**Weaknesses:**

The design choices are mostly experience-guided: e.g. identifying the projection layer as the ArtiHippo, using the mean class tokens to measure task similarity, and four operations to grow the ArtiHippo. More discussions on principles and analysis would make the paper more solid.

**Questions:**

1. Could some evaluation on the soundness of measuring task-similarity with the normalized cosine similarity between the mean class tokens be provided?
2. Would the same conclusion hold for stronger/larger ViT models?

---

> ### Author Response · Authors · 2023-11-22
> **Rebuttal by Authors**
>
> Thank you for your time and efforts reviewing our paper. We address your concerns as follows, which will be carefully updated in the revision.
>
> ### The design choices are mostly experience-guided: e.g. identifying the projection layer as the ArtiHippo, using the mean class tokens to measure task similarity, and four operations to grow the ArtiHippo. More discussions on principles and analysis would make the paper more solid.
>
> Principles and Analysis: We agree that principles and analysis would be beneficial, we would like to note that a theoretical analysis of Transformer models is largely an open area of research. [1] provide some theoretical analysis of the role played by each component of transformers, but are restricted to toy problems. Moreover, such a theoretical analysis of Transformers applied to images is lacking. We agree that this is certainly an important area that needs to be investigated, but leave it for future work for a comprehensive treatment.
>
> ###  Could some evaluation on the soundness of measuring task-similarity with the normalized cosine similarity between the mean class tokens be provided?
>
>  Our choice of the use of class token is similar to Learn to Prompt, which uses the class token from the last layer to measure the similarity between the keys of the prompt pool and the current image. Following Learn to Prompt, we too use the cosine similarity to measure task similarity.
>
> ### Would the same conclusion hold for stronger/larger ViT models?
>
>  Due to resource constraints, we have left this for future work. However, some work from Parameter Efficient Fine-Tuning [2] suggests that our conclusions might hold for larger models as well. In particular, Figure 2 in [2] suggests that methods that tune the parameters (Compacter, Compacter++ [3], IA3 [2], LoRA [4], etc.) perform better than prompt based approaches. We could expect the same trend to follow here as well.
>
> > [1] 	Alberto Bietti, Vivien Cabannes, Diane Bouchacourt, Hervé Jégou, Léon Bottou: Birth of a Transformer: A Memory Viewpoint. NeurIPS 2023.
>
> > [2] Haokun Liu, Derek Tam, Mohammed Muqeeth, Jay Mohta, Tenghao Huang, Mohit Bansal, Colin Raffel: Few-Shot Parameter-Efficient Fine-Tuning is Better and Cheaper than In-Context Learning. NeurIPS 2022
>
> > [3] Rabeeh Karimi Mahabadi, James Henderson, Sebastian Ruder: Compacter: Efficient Low-Rank Hypercomplex Adapter Layers. NeurIPS 2021: 1022-1035
>
> > [4] Edward J. Hu, Yelong Shen, Phillip Wallis, Zeyuan Allen-Zhu, Yuanzhi Li, Shean Wang, Lu Wang, Weizhu Chen: LoRA: Low-Rank Adaptation of Large Language Models. ICLR 2022

---

> > ### Comment · Reviewer_6RhY · 2023-11-23
> > **Thanks for the response**
> >
> > I've read the reviewers' comments and the authors' responses. I agree that the novelty of this work is under question, but I also reckon with the authors that applying existing techniques to new problems is also important: the proposed approach is thoroughly evaluated on challenging benchmarks showing consistent improvements. Therefore I'll keep my original score.

---

### Official Review · Reviewer_UTmX · 2023-11-06

**Soundness:** 3 good
**Presentation:** 3 good
**Contribution:** 1 poor
**Rating:** 5
**Confidence:** 4

**Summary:**

In this paper, the authors tackle the problem of lifelong learning in deep neural networks, specifically focusing on Vision Transformers (ViTs). They introduce a concept inspired by the human brain's hippocampus, called Artificial Hippocampi (ArtiHippo), to help ViTs learn continuously without forgetting previous knowledge—a common challenge known as catastrophic forgetting. The study explores where to place ArtiHippo within ViTs, what kind of structure it should have, and how it can grow and adapt over time while retaining past knowledge. By testing their approach on challenging benchmarks, the authors demonstrate that their method not only performs better than previous ones but also marks the first successful application of lifelong learning in Vision Transformers, showing great promise for future AI systems.

**Strengths:**

1. Balanced Exploration and Exploitation: The proposed method offers a new searching strategy that balances exploration (learning new information) and exploitation (using existing knowledge), crucial for the development of robust lifelong learning systems.

2. Empirical Results: The approach is thoroughly evaluated on challenging benchmarks, where it consistently outperforms existing methods. This demonstrates the practical effectiveness of the proposed solution and its potential for real-world applications.

**Weaknesses:**

1. Refinement Rather Than Revolution: The integration of ArtiHippo into Vision Transformers, though presented as a novel idea, is actually a clever twist on the established "learning to grow" concept. It's a smart update, but it falls short of being a game-changer. It feels like we're seeing a refinement of existing ideas rather than a bold reimagining of lifelong learning.

2. A Safe Bet Over a Leap of Faith: Employing Reuse, New, Adapt, and Skip operations within Transformers comes across as a safe, almost expected move. It's as if the paper takes a well-trodden path, applying tried-and-tested strategies to new territory, rather than venturing into unexplored innovative realms.

3. Narrow Lens on Competing Approaches: The paper misses a beat by not sizing up ArtiHippo against the full spectrum of lifelong learning strategies, particularly gradient-based and regularization-based methods. This omission leaves us guessing about how ArtiHippo truly stacks up against the competition and muddies the waters of its potential as a standout solution in the field.

**Questions:**

In what ways does the ArtiHippo framework conceptually and functionally diverge from the 'Learning to Grow' methodology, considering the apparent similarities in their approach to lifelong learning?

---

> ### Author Response · Authors · 2023-11-22
> **Rebuttal by Authors**
>
> Thank you for your time and efforts reviewing our paper. We address your concerns as follows, which we will carefully update in the revision.
>
> ### Summary: ... By testing their approach on challenging benchmarks, the authors demonstrate that their method not only performs better than previous ones but also marks the first successful application of lifelong learning in Vision Transformers, showing great promise for future AI systems.
>
> Thank you for your positive comments. We would like to point out that our proposed method is actually not the first work to apply Vision Transformers for lifelong learning. We have provided a review of prior work that uses ViTs in a lifelong learning setting in Section 2. However, we are indeed the first to evaluate ViTs in a lifelong learning setting which contains very diverse tasks, i.e., the VDD benchmark as mentioned in the Abstract.
>
> ### Refinement Rather Than Revolution
>
> While our method is built on Learn to Grow and SPOS, we make the following novel contributions, and also show the limitations of the two methods:
>
> 1. We show that the choice of where to apply NAS matters for achieving good forward transfer. We propose to use the final Linear projection layer in the MHSA block, which is a very lightweight component. Through ablation studies, we validate that this choice indeed results in the best average accuracy. Please see Table 6 in the Appendix for a comparison with other components (Value, Query, Key, and FFN). While it has been shown that finetuning the entire MHSA block achieves good downstream performance [1], we show that using only the projection layer is an equally viable (and more efficient) option.
> 2. We show that the original Learn to Grow which uses DARTS on ConvNets cannot perform well when applied to ViTs, even when an advanced version of DARTS ($\beta$-DARTS) is used (Rows 4 and 5 in Table 2).
> 3. We show that using the original SPOS formulation, which samples all the operators uniformly, is not sufficient to achieve better performance than DARTS (Row 6 in Table 2).
> 4. We propose a novel algorithm to convert task similarities (as measured by the normalized cosine similarity between the mean class tokens of each task) into a prior distribution over the NAS operations (reuse, adapt, new, skip). Intuitively, this translates to assigning higher probability to the reuse operation if the mean class token calculated using the data of task is similar to the mean class token calculated using the data of task .
> 5. We propose an exploration-exploitation driven sampling strategy, which is designed to enhance the prior. Here, exploration refers to sampling from a uniform distribution over the operations, and exploitation refers to sampling from the prior distribution calculated in Eqn.3. We show that this strategy significantly outperforms the origin Learn to Grow, as well as popular prompt based approaches.
> 6. We further show that our strategy is complementary to the prompt based approaches, and combining them leads to even higher performance.
>
> > [1] Hugo Touvron, Matthieu Cord, Alaaeldin El-Nouby, Jakob Verbeek, Hervé Jégou: Three Things Everyone Should Know About Vision Transformers. ECCV (24) 2022: 497-515
>
> ### A Safe Bet Over a Leap of Faith
>
> While the use of 4 growing operations Reuse, New, Adapt, and Skip has been explored before, we are the first to apply this paradigm to Vision Transformers. However, ViTs are computationally expensive class of models, and applying the operations uniformly to all the layers will be extremely costly. We propose, and empirically validate the choice of the linear projection layer in the MHSA block to apply the 4 operations effectively.
>
> Venturing into unexplored innovative realms is certainly important, but applying existing techniques to new problems is also equally important. As we show in Table 2 in the main text, applying existing techniques to new problems is not straightforward, as evidenced by the low average accuracy of Learn to Grow in its original form (rows 4 and 5), and SOPS in its original form (row 6). Hence, our contribution also lies in addressing the weaknesses of the existing methods when applied to ViTs, and proposing novel techniques to overcome them (please see our response to (2)).
>
> ### Narrow Lens on Competing Approaches
>
> Could you please clarify the meaning of/provide references to gradient-based approaches in lifelong learning?
> While not a completely fair comparison, we compare with replay-based approaches and regularization-based approaches (Table 7 in the Appendix) to inform readers of the tradeoff of using these methods. Note that replay-based and regularization-based methods suffer from catastrophic forgetting, hence requiring the storage of all the previous models.

---

> > ### Comment · Reviewer_UTmX · 2023-11-22
> > **Thanks for the response**
> >
> > I am not full convinced about the claimed novelty, since, all the searching strategy and designed operation for grow, are already there. Simply adding more components to the system doesn't fundamentally alter its essence; there's no significant conceptual change. The system doesn't offer any capabilities beyond what 'Learn to Grow' already does.
> >
> > The use of 'Learn to Grow' in transformers, including example [A], is not entirely novel. Additionally, if expanding the 'Learn to Grow' concept to include the addition of new learnable prompts, there is a wealth of related work in this area.
> >
> > However, I appreciate the thorough evaluations provided.
> >
> > Therefore, I will keep my original score of 5. Hope the author can understand my concern.
> >
> > [A] Gao, Qiankun, et al. "A Unified Continual Learning Framework with General Parameter-Efficient Tuning." ICCV 2023

---

> > > ### Author Response · Authors · 2023-11-22
> > > **Thank you for your quick response**
> > >
> > > Dear UTmX,
> > >
> > > We understood your concerns, but we think there are a few aspects worthing to be clarified further.
> > >
> > > ### The system doesn't offer any capabilities beyond what 'Learn to Grow' already does.
> > >
> > > > We clearly show the vanilla 'Learn to Grow' (L2G) does not work well using ViTs, even with more powerful $\beta$-DARTS methods. The vanilla L2G does not explore where to apply the growing at all. We addresses both in the submission with significantly better performance obtained.
> > >
> > > ### For the new reference [A]
> > >
> > > > Thank you for point it out. We will address it in the revision. After looking into the paper, we appreciate you for sharing the excellent work.  The main components of the proposed LAE are  *"1) Learning: the pre-trained model adapts to the new task by tuning an online PET module, along with our adaptation speed calibration to align different PET modules, 2) Accumulation: the task-specific knowledge learned by the online PET module is accumulated into an offline PET module through momentum update, 3) Ensemble: During inference, we respectively construct two experts with online/offline PET modules (which are favored by the novel/historical tasks) for prediction ensemble."*
> > >
> > > > With all due respects, if we understood it correctly,  does the proposed LAE method in [A] have similar situations like ours per your comments, "all the searching strategy and designed operation for grow, are already there. Simply adding more components to the system doesn't fundamentally alter its essence; there's no significant conceptual change."
> > >
> > > > We appreciate the excellent work of [A].  We would also like to request you, if possible,  to re-consider the technically novel components and the resulted performance improvement in our proposed ArtiHippo.

---

### Official Review · Reviewer_QBWR · 2023-11-06

**Soundness:** 3 good
**Presentation:** 4 excellent
**Contribution:** 2 fair
**Rating:** 5
**Confidence:** 2

**Summary:**

This paper addresses the challenge of catastrophic forgetting in deep neural networks, particularly focusing on Transformer models, which are increasingly popular but complex to train for lifelong learning tasks. It introduces a concept called Artificial Hippocampi (ArtiHippo) within Vision Transformers (ViTs) to enhance their resiliency and adaptability for continual learning, drawing inspiration from the human brain and demonstrating improved performance on rigorous benchmarks.

**Strengths:**

+ The paper is well written, and the idea is also elaborated clearly. There is no trouble in reading and reproducing the algorithm.
+ In addition, the implementation and code are provided to demonstrate the proposed algorithm.
+ Experiments provide adequate support to the assumptions and claims and prove that the new method is applicable to vision datasets.

**Weaknesses:**

- The major concern is the lack of novelty for the paper to be published in ICLR. Although the paper is well thought out and demonstrated with extensive experiments, the novelty is rather incremental. See the question section.
- Another issue is the research problem “task incremental learning,” which seems not very practical in many vision applications. As the authors discussed, “class incremental” is more demanded since the task information is not always (unlikely) available.
- Some comparisons in experiments seem unfair and may not be able to reflect the true performance of each framework.

**Questions:**

1.	The key methodology of this paper is mainly built upon several existing works. The four operations are adapted from “learning-to-grow,” and the construction of supernet is based on SPOS. Although finding the right place to place ArtiHippo is tricky, it has been common sense that classification information is mostly likely located on the upper level of the network.
2.	In addition, the method leverages the recent “prompt” based method such as S-Prompts to further improve the performance. This makes the neat “ArtiHippo” based algorithm a bit over-complicated and more engineering-driven.
3.	It seems all methods always use pre-trained strong ViT backbone for incremental tasks. This does cover all the cases in lifelong learning, e.g., starting from a moderated size model, or from zero knowledge. The current methodology is still like finetuning a strong baseline on several small datasets. The problem discussed and experiments are not typical lifelong learning scenarios.
4.	The major concern lies in the knowledge of “task.” It does not always make sense to know where the data are from or their sources.

---

> ### Author Response · Authors · 2023-11-22
> **Rebuttal by Authors (1/2)**
>
> Thank you for your time and efforts reviewing our paper. We address your concerns as follows, which will be carefully updated in the revision.
>
> ### Lack of novelty and similarity with Learn to Grow and SPOS.
>
> While our method is built on Learn to Grow and SPOS, we make the following novel contributions, and also show the limitations of the two methods:
> 1. We show that the choice of where to apply NAS matters for achieving good forward transfer. We propose to use the final Linear projection layer in the MHSA block, which is a very lightweight component. Through ablation studies, we validate that this choice indeed results in the best average accuracy. Please see **Table 6 in the Appendix** for a comparison with other components (Value, Query, Key, and FFN). While it has been shown that finetuning the entire MHSA block achieves good downstream performance [1], we show that using only the projection layer is an equally viable (and more efficient) option.
> 2. We show that the original Learn to Grow which uses DARTS on ConvNets cannot perform well when applied to ViTs, even when an advanced version of DARTS ($\beta$-DARTS) is used (**Rows 4 and 5 in Table 2**).
> 3. We show that using the original SPOS formulation, which samples all the operators uniformly, is not sufficient to achieve better performance than DARTS (**Row 6 in Table 2**).
> 4. We propose a novel algorithm to convert task similarities (as measured by the normalized cosine similarity between the mean class tokens of each task) into a prior distribution over the NAS operations (reuse, adapt, new, skip). Intuitively, this translates to assigning higher probability to the reuse operation if the mean class token calculated using the data of task $t-1$ is similar to the mean class token calculated using the data of task $t$.
> 5. We propose an exploration-exploitation driven sampling strategy, which is designed to enhance the prior. Here, **exploration** refers to sampling from a uniform distribution over the operations, and **exploitation** refers to sampling from the prior distribution calculated in Eqn.3. We show that this strategy significantly outperforms the origin Learn to Grow, as well as popular prompt based approaches.
> 6. We further show that our strategy is complementary to the prompt based approaches, and combining them leads to even higher performance.
>
> > [1] Hugo Touvron, Matthieu Cord, Alaaeldin El-Nouby, Jakob Verbeek, Hervé Jégou: Three Things Everyone Should Know About Vision Transformers. ECCV (24) 2022: 497-515
>
>
> ### Although finding the right place to place ArtiHippo is tricky, it has been common sense that classification information is mostly likely located on the upper level of the network.
>
> We respectfully disagree with this statement. [2] show that which layer to finetune depends on the nature of the data, and surgically choosing which layer to finetune outperforms tuning the last layer. Consequently, which layer to reuse, adapt, renew or skip in the learn to grow will depend on the nature of the current task, and our method provides a framework to derive a prior distribution on the four operators tased on task similarities, rather than manually inspecting which layer to tune for each incoming task.
>
> > [2] Yoonho Lee, Annie S. Chen, Fahim Tajwar, Ananya Kumar, Huaxiu Yao, Percy Liang, Chelsea Finn: Surgical Fine-Tuning Improves Adaptation to Distribution Shifts. ICLR 2023
>
> ### Another issue is the research problem “task incremental learning,” which seems not very practical in many vision applications
>
> We acknowledge that the task incremental setting is a limitation of our method.  We note that the class incremental setting has its own limitations, e.g., it often requires the number of classes between different tasks is the same, which is not applicable in the VDD benchmark we tested. In the meanwhile,  the task incremental  setting remains a commonly used setting in studying lifelong learning, where  the focus is how to achieve maximal forward transfer across tasks that may have different output space  by reusing the relevant parameters learned for all the previous tasks and adding minimal new parameters, which is also an equally important problem.
>
> ### Some comparisons in experiments seem unfair and may not be able to reflect the true performance of each framework.
>
> Please note that we modify all the methods to work in a task incremental setting for a fair comparison with our method. We request you to elaborate which comparisons seem unfair, so that we can address your concerns in a comprehensive manner.

---

> ### Author Response · Authors · 2023-11-22
> **Rebuttal by Authors (2/2)**
>
> ### In addition, the method leverages the recent “prompt” based method such as S-Prompts to further improve the performance. This makes the neat “ArtiHippo” based algorithm a bit over-complicated and more engineering-driven.
>
> We would like to clarify that integration with S-Prompts does not form the core component of our proposed method, but rather an initial exploration of integration of the two approaches due to their complementary behavior. As shown in Table 2, rows 7 and 8, our proposed method without S-Prompt performs better than Learn to Prompt$^\dagger$, S-Prompts$^\dagger$ and Learn to Grow. The integration with S-Prompt, although very simple, **further** boosts the average accuracy. We leave a more comprehensive (and possibly simpler) integration for future work.
>
> ### It seems all methods always use pre-trained strong ViT backbone for incremental tasks. ... The problem discussed and experiments are not typical lifelong learning scenarios.
>
> We agree that this is not a typical lifelong learning scenario, but a different and equally important scenario of continually improving a pretrained model. This methodology has been used in all the works we compare with, and is the fundamental idea behind prompt based methods, i.e., Learning to Prompt (CVPR22), S-Prompts (NeurIPS22), as well as Efficient Feature Transformation (CVPR21), and Lightweight Learner (TMLR23). Hence, the comparisons with these methods are fair. We show that adding plasticity through model parameters achieves better forward transfer than all these methods (as evidenced by the average accuracy Table 2 and Table 4). Please see the advantages of our method in Sections 4.1 and 4.4.
>
> ### Justifying the selection of the projection layer in the MHSA
>
> To further show and verify the effectiveness of the selection of the projection layer in MHSA, we conduct another experiments of fine-tuning the ImageNet-1k trained ViT-B backbone on the fine-grained visual classification (FGVC) benchmark consisting of five categories using the popoular LoRA method [3].  In the literature, LoRA is often applied the Query/Key/Value of Transformers. Our preliminary results show that applying LoRA to the projection layer in the MHSA is much more effective as shown in the table below, which shows that the selection of the projection layer is non-trivial and has a good potential.
>
> | Method | CUBS [4] | Birds [5] | Flower [6] | Dog [7] | Car [8] | Avg|
> | :---: | :---: | :---: | :---: | :---: | :---: | :---: |
> | LoRA (QKV) | 82.5 | 71.2 | 81.2 | 97.5 | 76.6 | 81.8 |
> | LoRA (Proj) | 81.8 | 76.3 | 92.5 | 97.8 | 81.4 | 86.0 |
>
>
> > [3] Edward J Hu, yelong shen, Phillip Wallis, Zeyuan Allen Zhu, Yuanzhi Li, Shean Wang, Lu Wang, and Weizhu Chen. LoRA: Low-rank adaptation of large language models. ICLR, 2022
>
> > [4] Catherine Wah, Steve Branson, Peter Welinder, Pietro Perona, and Serge Belongie. The Caltech-UCSD Birds-200-2011 Dataset. 2011
>
> > [5] Grant Van Horn, Steve Branson, Ryan Farrell, Scott Haber, Jessie Barry, Panos Ipeirotis, Pietro Perona, and Serge J. Belongie. Building a bird recognition app and large scale dataset with citizen scientists: The fine print in fine-grained dataset collection. CVPR 2015
>
> > [6] Maria-Elena Nilsback and Andrew Zisserman. Automated flower classification over a large number of classes. In Sixth Indian Conference on Computer Vision, Graphics & Image Processing, 2008
>
> > [7] Timnit Gebru, Jonathan Krause, Yilun Wang, Duyun Chen, Jia Deng, and Li Fei-Fei. Fine-grained car detection for visual census estimation, AAAI 2017
>
> > [8] Aditya Khosla, Nityananda Jayadevaprakash, Bangpeng Yao, and Li Fei-Fei. Novel dataset for fine-grained image categorization. CVPRW 2011

---

### Comment · Area_Chair_YJvK · 2023-11-22
**Less than a day remaining**

Dear Reviewers,

If you have already responded to authors last response, Thank you!
If not, please take some time, read their responses and acknowledge by replying to the comment. Please also update your score, if applicable.

Thanks everyone for a fruitful, constructive, and respectful review process.

Cheers, Your AC!

---

### Author Response · Authors · 2023-11-22
**Global Response**

We sincerely thank the reviewers for their constructive feedback which will help us to greatly improve our submission. We appreciate the ACs for their valuable time spent discussing our submission. We address the reviewers’ comments individually.

We have revised the PDF to include the following:

### Additional references: We have added 4 more references to the literature review:

> [1] Zhen Wang, Liu Liu, Yiqun Duan, Yajing Kong, Dacheng Tao: Continual Learning with Lifelong Vision Transformer. CVPR 2022: 171-181

> [2] Abdelrahman Mohamed, Rushali Grandhe, K. J. Joseph, Salman H. Khan, Fahad Shahbaz Khan: D3Former: Debiased Dual Distilled Transformer for Incremental Learning. CVPR Workshops 2023: 2421-2430

> [3] James Seale Smith, Leonid Karlinsky, Vyshnavi Gutta, Paola Cascante-Bonilla, Donghyun Kim, Assaf Arbelle, Rameswar Panda, Rogério Feris, Zsolt Kira: CODA-Prompt: COntinual Decomposed Attention-Based Prompting for Rehearsal-Free Continual Learning. CVPR 2023: 11909-11919

> [4] Qiankun Gao, Chen Zhao, Yifan Sun, Teng Xi, Gang Zhang, Bernard Ghanem, and Jian Zhang. A unified continual learning framework with general parameter-efficient tuning. ICCV 2023: 11483-11493

### We have added a section in the Appendix (Section E) discussing the previous literature on using Vision Transformers for lifelong learning, and how it differs from our work.

---

### Meta-Review · Area_Chair_YJvK · 2023-12-09

**Metareview:**

This paper proposes a methd to enable lifelong learning in VT without catastrophic forgetting. The proposed method identifies the final projection layer in multi-head self-attention blocks as a lightweight component to dynamically grow using neural architecture search guided by task similarities and performed with 4 operations. Experiments on visual decathlon and 5-dataset benchmarks show the method achieves better performance than prior ones.

Strengths:
- The paper is clearly written and well visualied and easy to follow.
- The proposed method seems to be effective at enabling lifelong learning in VTs, outperforming prior ones on the benchmarks tested. The analysis of results is good and insightful.
- The concept of Artificial Hippocampi in Transformers is interesting for tackling task-incremental learning. Moreover, introduces lightweight and adaptable components to handle new tasks.
- The proposed exploration-exploitation search strategy balances learning new information and exploiting existing knowledge well. This is crucial for lifelong learning systems. It coincdes well with stability dilemma well known in continual leanring literature.


Weaknesses:
- The design choices like selecting the projecton layer, using class tokens for similarity, are experience driven. This limits the novelty and impact of the paper..
- The method relies on knowing the task identity at test time, limiting applicability. Scaling to a larger number of tasks and model sizes is not analyzed.
- Extending the scope and applicablity of the work, with more justification or intution on the decisions made will help the paper a lot for future submissions.

**Justification For Why Not Higher Score:**

lack of solid contribution and limited scope and applicability of the result inhibits me from accepting this paper.

**Justification For Why Not Lower Score:**

NA

---

### Decision · Program_Chairs · 2024-01-16

Reject